# MVR: Multi-view Video Reward Shaping for Reinforcement Learning

**Lirui Luo**[1,2,*], **Guoxi Zhang**[2,*], **Hongming Xu**[2], **Yaodong Yang**[1,2], **Cong Fang**[1,3,†], **Qing Li**[2,†]

[1] State Key Lab of General AI, School of Intelligence Science and Technology, Peking University
[2] State Key Laboratory of General Artificial Intelligence, BIGAI
[3] Institute for Artificial Intelligence, Peking University
fangcong@pku.edu.cn, dylan.liqing@gmail.com
Project page: https://mvr-rl.github.io/

## Abstract

Reward design is of great importance for solving complex tasks with reinforcement learning. Recent studies have explored using image-text similarity produced by vision-language models (VLMs) to augment rewards of a task with visual feedback. A common practice linearly adds VLM scores to task or success rewards without explicit shaping, potentially altering the optimal policy. Moreover, such approaches, often relying on single static images, struggle with tasks whose desired behavior involves complex, dynamic motions spanning multiple visually different states. Furthermore, single viewpoints can occlude critical aspects of an agent's behavior. To address these issues, this paper presents Multi-View Video Reward Shaping (MVR), a framework that models the relevance of states regarding the target task using videos captured from multiple viewpoints. MVR leverages video-text similarity from a frozen pre-trained VLM to learn a state relevance function that mitigates the bias towards specific static poses inherent in image-based methods. Additionally, we introduce a state-dependent reward shaping formulation that integrates task-specific rewards and VLM-based guidance, automatically reducing the influence of VLM guidance once the desired motion pattern is achieved. We confirm the efficacy of the proposed framework with extensive experiments on challenging humanoid locomotion tasks from HumanoidBench and manipulation tasks from MetaWorld, verifying the design choices through ablation studies.

## 1 Introduction

When using reinforcement learning (RL), agents learn to optimize *task rewards* encoding quantitative objectives. For example, in the running task shown in Fig. 1, the task rewards incentivize the tracking of a designated forward speed. In contrast, when learning new skills, humans can benefit from visual guidance tailored to the desired motion pattern, thus preventing suboptimal postures. Inspired by this observation, we study the incorporation of visual feedback into task rewards in this paper.

With the development of Vision-Language Model (VLM)s, an emerging paradigm is to leverage image-text similarity as rewards Chan et al. (2023); Rocamonde et al. (2024), which guides agents to visit states that best match the textual description of a task. However, this paradigm faces three shortcomings. Firstly, as single-image similarity cannot characterize dynamic motions, the generated rewards are mis-specified—they incentivize agents to visit the very states with maximal similarity score (Fu et al., 2024). As a result, agents cannot learn rhythmic movements involving repeated transitions between different states. Moreover, single viewpoints suffer from occlusion between robot limbs, creating view-dependent bias that further destabilizes dense guidance. Selecting the viewpoint with minimal occlusion often requires extensive human effort and domain knowledge. In addition, most methods linearly add single-view VLM scores to task or success rewards without explicit shaping, which can alter the optimal policy.

---

[*]Equal contribution.
[†]Corresponding authors.

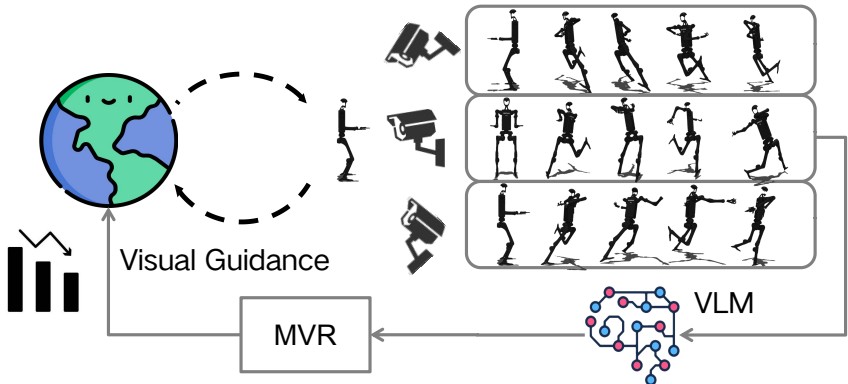

Figure 1: **The proposed MVR computes visual guidance using a VLM and videos collected from multiple viewpoints.** In this example, the task requires a humanoid robot to run forward. Being captured from different viewpoints, the image sequences encode complementary information and enable comprehensive evaluation of the agent's behaviors. This example also illustrates the pitfall of using image-text similarity for dynamic tasks—running requires rhythmic alternation of legs, but optimizing image-text similarity leads to realizing the pose that best matches "running" repeatedly. The shaping term prioritizes states that establish alternating leg cadence; once a stable gait and the target forward speed are achieved, its influence automatically decreases and the task reward takes precedence (see Sec. 4).

This paper presents Multi-view Video Reward Shaping (MVR), an online RL framework that overcomes these issues. As illustrated in Fig. 1, MVR collects multi-view videos of agent behaviors and estimates state relevance from video-text similarity. This learned relevance constructs a *state-dependent reward shaping* term that provides dense guidance early on and automatically diminishes as behavior improves, avoiding persistent conflict with task objectives.

We evaluate the effectiveness of MVR on a total of 19 tasks from HumanoidBench (Sferrazza et al., 2024) and MetaWorld (Yu et al., 2019). MVR consistently outperforms VLM-RM (Rocamonde et al., 2024), and RoboCLIP (Sontakke et al., 2024), which are existing VLM-based methods. Furthermore, we validate the design of MVR in an ablation study and investigate the influence of using multiple views. Lastly, we showcase the effectiveness of visual feedback for preventing suboptimal behaviors. The contributions of this paper are as follows.

1. We propose MVR, a framework for generating feedback from multi-view videos.

2. We introduce a *state-dependent reward shaping* formulation that integrates task rewards with VLM-based guidance and decays its influence as behaviour aligns with the target.

3. We validate MVR's effectiveness through simulated experiments on humanoid locomotion tasks in HumanoidBench and manipulation tasks in MetaWorld, and present a case study to showcase the importance of visual guidance.

## 2 RELATED WORK

**VLM-based and Video-based Rewards**    Recent work leverages VLMs to generate reward signals through image-text similarity (Chan et al., 2023; Rocamonde et al., 2024; Kim et al., 2023; Cui et al., 2024) or goal images (Gao et al., 2023). These methods typically treat the VLM as a frozen scoring function and then improve reward quality via VLM fine-tuning (Fu et al., 2024), binarization (Huang et al., 2024a), or ranking-based objectives (Wang et al., 2024b).

Beyond single-frame similarity, a growing body of work learns rewards directly from videos of task executions. Rank2Reward (Yang et al., 2024) combines a temporal ranking-based reward model with an adversarial (GAIL-style) discriminator; PROGRESSOR (Ayalew et al., 2024) learns a progress estimator and applies reward-weighted regression on real robots; and GVL (Ma et al., 2025) uses a Gemini-based VLM to provide zero-shot progression rewards for downstream applications. Generative video models such as Diffusion Reward (Huang et al., 2024b) and GenFlowRL (Yu

et al., 2025) define rewards from conditional video diffusion or object-centric flow, while Video-Language Critic (VLC) (Alakuijala et al., 2025) fine-tunes a CLIP4Clip-style backbone so that a video-language critic can provide transferable rewards for training language-conditioned RL policies. REDS (Kim et al., 2025) learns subtask-aware dense rewards from segmented demonstrations, and ReWiND (Zhang et al., 2025) trains a video-language transformer to produce dense language-conditioned rewards and demonstrates real-world RL fine-tuning.

However, most of these approaches define rewards directly on raw image or video observations, without learning an explicit state-space relevance function, or assume access to curated demonstration datasets. Classical example-guided controllers such as DeepMimic (Peng et al., 2018) and recent state-only imitation–emulation frameworks such as CIMER (Han et al., 2024) also rely on expert trajectories rather than re-using unlabeled online rollouts. RoboCLIP (Sontakke et al., 2024) is closer to our setting and uses video-text similarity but provides only sparse trajectory-level rewards. MVR differs by learning state relevance from multi-view videos through paired comparisons and introducing state-dependent shaping that automatically decays, providing dense guidance during online RL while remaining compatible with task rewards.

**RL with Foundation Models** Foundation models have been extensively integrated into RL pipelines (Xu et al., 2024; Moroncelli et al., 2024; Kawaharazuka et al., 2024). VLMs enable planning (Pan et al., 2024; Patel et al., 2023), success detection (Du et al., 2023a), and representation learning (Chen et al., 2024), though most rely on single-frame inputs unsuitable for dynamic tasks. LLMs facilitate planning (brian ichter et al., 2022; Shinn et al., 2024), exploration (Du et al., 2023b), code-based policy generation (Liang et al., 2023), and reward design (Ma et al., 2024; Xie et al., 2024). World models provide alternative approaches for reward generation (Escontrela et al., 2023) and model-based RL (Wu et al., 2024). Our work specifically addresses the unique challenges of using pre-trained VLMs for continuous motion tasks, proposing practical solutions for video-based multi-view reward learning.

## 3 PROBLEM STATEMENT

**Markov Decision Process** The Markov Decision Process (MDP) Sutton (2018) is defined as $\langle \mathcal{S}, \mathcal{A}, P, r, \gamma \rangle$, where $\mathcal{S}$ and $\mathcal{A}$ denote the state space and the action space. $P : \mathcal{S} \times \mathcal{A} \to \Delta(\mathcal{S})$ is the transition probability of states, and $r : \mathcal{S} \times \mathcal{A} \to \mathbb{R}$ is the reward function. $\gamma \in (0, 1)$ is the discount factor. An MDP prescribes an interaction protocol in which an agent repeatedly receives states from an environment, samples actions from its policy $\pi(a|s) : \mathcal{S} \to \Delta(A)$, and receives rewards. The value function $v^{\pi}(s) = \mathbb{E}\left[\sum_{t=1}^{\infty} \gamma^{t-1} r_t\right]$ characterizes the expected discounted return of $\pi$ starting from state $s$. The objective of RL is to find the optimal policy $\pi^*$ such that $v^{\pi^*}(s) \geq v^{\pi}(s)$ for any state $s$.

**The Learning Problem** The video-text similarity computed by models such as ViCLIP (Wang et al., 2024a) is denoted by $\psi^{\text{VLM}}$ and characterizes how well a video matches a text description. During policy learning, we periodically sample state sequences $\mathbf{s}$, render them into videos $\mathbf{o}$ from different viewpoints, and compute the video-text similarity scores $\psi^{\text{VLM}}(\mathbf{o}, \ell)$, where $\ell$ is the task description. For notational convenience, we also use $\psi^{\text{VLM}}(\mathbf{o}, \mathbf{o}')$ for the similarity between two videos $\mathbf{o}$ and $\mathbf{o}'$. These steps result in a dataset $\mathcal{D} = \{(\mathbf{s}, \mathbf{o}, \psi^{\text{VLM}}(\mathbf{o}, \ell))\}$. Throughout the paper, we use bold symbols (e.g., $\mathbf{s}$) for sequences and plain symbols (e.g., $s$) for single timesteps. Given $\mathcal{D}$, we aim to learn a *state relevance model* $f^{\text{MVR}} : \mathcal{S} \to \mathbb{R}$ with parameters $\theta^{\text{MVR}}$ to model the relevance of states for the task of interest. With $f^{\text{MVR}}$, we derive a reward function $r^{\text{VLM}} : \mathcal{S} \to \mathbb{R}$ as visual feedback for the agent, complementing the task rewards $r^{\text{task}}$.

## 4 MULTI-VIEW VIDEO REWARD SHAPING

The proposed MVR is illustrated in Fig. 2. Section 4.1 explains how the relevance model $f^{\text{MVR}}$ is learned, and Sec. 4.2 introduces the proposed reward function and the entire framework.

### 4.1 LEARNING STATE RELEVANCE FROM MULTI-VIEW VIDEOS

**Challenges**    Operating in state space $\mathcal{S}$ enables efficient reward computation without per-step video rendering, which is crucial for million-step training. However, this design raises two coupled challenges. First, there is a semantic gap between states (proprioception, joint angles) and visual features (textures, colors), which prohibits directly regressing video-text similarity from states. Second, MVR renders videos from one of four viewpoints ($0°$, $90°$, $180°$, and $270°$), which introduces systematic viewpoint biases: for instance, frontal views in Fig. 1 often yield higher similarities due to better visibility, creating spurious correlations unrelated to behavioral quality. In the remainder of this subsection, we address the semantic gap with *Matching Paired Comparisons* and the viewpoint-induced bias with *Regularizing State Representations*.

**Matching Paired Comparisons**    To tackle the semantic gap between states and videos, we propose to match comparisons in the two domains instead of regressing raw similarity scores. Being trained with contrastive objectives, the video-text similarity scores of two videos reflect the probability that one video better matches the task than the other. *We claim that such orderings should be preserved between the corresponding state sequences.* Specifically, the probability that a video $\mathbf{o}$ better matches the task prompt $\ell$ than another video $\mathbf{o}'$ under the Bradley-Terry (BT) model (Bradley & Terry, 1952) is given by

$$h_{\text{vid}}(\mathbf{o}, \mathbf{o}') = \sigma(\psi^{\text{VLM}}(\mathbf{o}, \ell) - \psi^{\text{VLM}}(\mathbf{o}', \ell)), \tag{1}$$

where $\sigma(x) = (1 + \exp(-x))^{-1}$ is the sigmoid function. Similarly, the probability that a state sequence $\mathbf{s}$ is better than another state sequence $\mathbf{s}'$ is

$$h_{\text{state}}(\mathbf{s}, \mathbf{s}') = \sigma\Big(\frac{1}{n(\mathbf{s})} \sum_{s \in \mathbf{s}} f^{\text{MVR}}(s) - \frac{1}{n(\mathbf{s}')} \sum_{s \in \mathbf{s}'} f^{\text{MVR}}(s)\Big), \tag{2}$$

where $n(\mathbf{s})$ is the length of $\mathbf{s}$. Given two samples $(\mathbf{s}, \mathbf{o}, \psi^{\text{VLM}}(\mathbf{o}, \ell))$ and $(\mathbf{s}', \mathbf{o}', \psi^{\text{VLM}}(\mathbf{o}', \ell))$, to preserve the ordering between videos and state sequences, we propose to minimize:

$$L_{\text{matching}} = -h_{\text{vid}}(\mathbf{o}, \mathbf{o}') \log(h_{\text{state}}(\mathbf{s}, \mathbf{s}')) - \big(1 - h_{\text{vid}}(\mathbf{o}, \mathbf{o}')\big) \log\big(1 - h_{\text{state}}(\mathbf{s}, \mathbf{s}')\big). \tag{3}$$

By minimizing this cross-entropy, $f^{\text{MVR}}$ can capture the relevance of states for the task without being distracted by the visual details. It is akin to the objective of preference-based RL Christiano et al. (2017), but with the crucial difference that we fit $f^{\text{MVR}}$ to probabilities rather than binary labels. Importantly, $(\mathbf{o}, \mathbf{o}')$ can originate from different viewpoints because each video shares the same underlying state sequence; sampling pairs across views enlarges the comparison dataset without requiring additional rendering.

**Regularizing State Representations**    The second challenge is the viewpoint-induced bias in $\psi^{\text{VLM}}$: different camera angles can change similarity scores for the same underlying behavior. A straightforward solution is to randomize the choice of viewpoint, but relevance learning then becomes difficult as we use more viewpoints, since we will have fewer samples from any single informative view. *To tackle this data scarcity while reducing viewpoint bias, we enhance relevance learning by regularizing state representations with the similarity structure between video embeddings.* Specifically, we parameterize $f^{\text{MVR}}$ as $f^{\text{MVR}}(s) = \langle g^{\text{rel}}, g^{\text{state}}(s) \rangle$, where $g^{\text{state}} : \mathcal{S} \to \mathbb{R}^d$ is a state encoder and $g^{\text{rel}} \in \mathbb{R}^d$ is a learnable vector. The scalar $d$ is the dimension of the state embeddings. This decomposition decouples representation learning from relevance scoring: $L_{\text{reg}}$ shapes a shared state representation, while $L_{\text{matching}}$ selects a single relevance direction $g^{\text{rel}}$ in that space, rather than entangling multi-view alignment and task-specific ranking in a single monolithic scalar predictor. Then, we use the following regularization term for relevance learning:

$$L_{\text{reg}} = \big| \psi^{\text{VLM}}(\mathbf{o}_i, \mathbf{o}_j) - \langle \bar{g}^{\text{state}}(\mathbf{s}_i), \bar{g}^{\text{state}}(\mathbf{s}_j) \rangle \big|, \tag{4}$$

where $\bar{g}^{\text{state}}(\mathbf{s}) = n(\mathbf{s})^{-1} \sum_{s \in \mathbf{s}} g^{\text{state}}(s)$ is the representation of a state sequence. This regularization leads to averaged state representations from which view-specific interference is reduced.

This regularization produces viewpoint-invariant state representations by aligning state and video similarity structures. While $L_{\text{matching}}$ preserves orderings, $L_{\text{reg}}$ anchors the learned embeddings, together stabilizing multi-view learning.

Finally, the overall objective for relevance learning is:

$$L_{\text{rel}} = L_{\text{matching}} + L_{\text{reg}}. \tag{5}$$

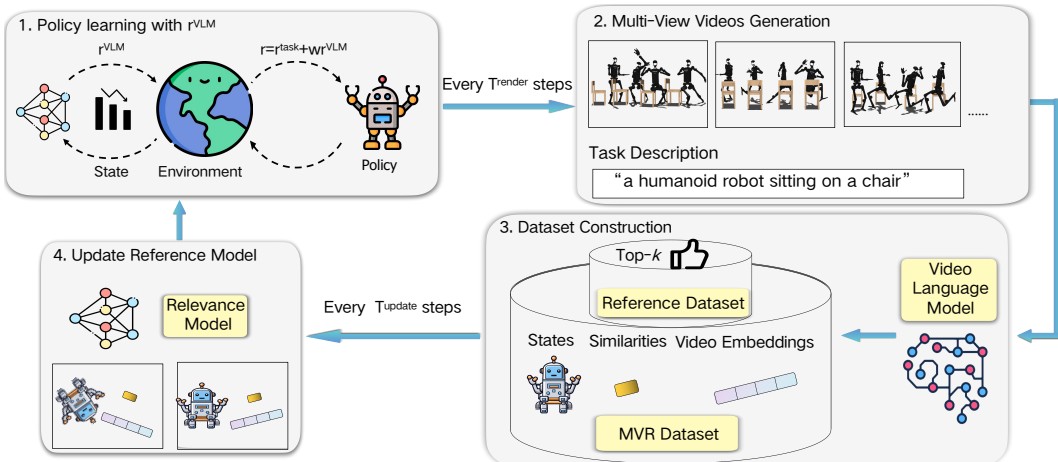

Figure 2: **The entire framework of the proposed MVR.** MVR periodically samples state sequences and renders them into videos from different viewpoints (step 2). It then queries a VLM for the similarity scores and video embeddings of the videos and augments its dataset $\mathcal{D}$ with state sequences, video embeddings, and similarity scores (step 3). Additionally, it keeps the state sequences with top-$k$ similarity scores in a reference set $\mathcal{D}^{\text{ref}}$. With $\mathcal{D}$, MVR updates the state relevance model (step 4). Lastly, using the latest state relevance model and the reference set, MVR computes visual feedback $r^{\text{VLM}}$ for the online RL agent (step 1), which is combined with task rewards $r^{\text{task}}$ through state-dependent reward shaping that automatically decays as the agent's behavior aligns with the reference set.

## 4.2 POLICY LEARNING WITH $r^{\text{VLM}}$

**Desiderata of $r^{\text{VLM}}$**    Similar to the case of image-text similarity, directly using state relevance as $r^{\text{VLM}}$ is prone to mis-specification. Instead, we opt for matching the expectation of state relevance–$r^{\text{VLM}}$ should lead to matching the learner's state distribution with videos of the highest video-text similarity, thus leading to the desired behaviors. Meanwhile, we argue that $r^{\text{task}}$ should gradually dominate $r^{\text{VLM}}$ as the learned behaviors resemble the desired one, which is crucial for avoiding the expensive VLM fine-tuning, especially when the VLM is not aligned with the target task.

**The Proposed Reward Function**    We first extend the notion of state relevance to policies.

**Definition 4.1.** Denote by $d^\pi(s) = (1-\gamma) \sum_{t=0}^\infty \gamma^t P(s_t{=}s \mid \pi)$ the state-occupancy distribution of policy $\pi$. The *policy relevance* $h^\pi$ is the expectation of state relevance under $d^\pi$:

$$h^\pi = \sum_{s \in \mathcal{S}} f^{\text{MVR}}(s) \, d^\pi(s).$$

Policy relevance characterizes the extent a policy matches the task. Denote by $\pi^\ell$ the policy that best matches the task prompt $\ell$. *Our idea is to learn policies that maximize $r^{task}$ while being indiscernible from $\pi^\ell$*:

$$\max_\pi v^\pi + w \log(\sigma(h^\pi - h^{\pi^\ell})), \tag{6}$$

where $\log(\sigma(h^\pi - h^{\pi^\ell}))$ is the log-likelihood of $\pi$ being better than $\pi^\ell$ under the BT model. $w \in \mathbb{R}$ is a weighting term. Using Jensen's inequality, this log-likelihood can be lower-bounded as:

$$\log \sigma(h^\pi - h^{\pi^\ell}) \geq \mathbb{E}_{s \sim d^\pi, s' \sim d^{\pi^\ell}} \left[ \log(\sigma(f^{\text{MVR}}(s) - f^{\text{MVR}}(s'))) \right]. \tag{7}$$

Thus, we can maximize Eq. (6) by using the shaped reward $r^{\text{MVR}}$ defined as

$$r^{\text{MVR}}(s) \triangleq r^{\text{task}}(s) + w r^{\text{VLM}}(s), \tag{8}$$

where $r^{\text{VLM}}$ is given by:

$$r^{\text{VLM}}(s) \triangleq \mathbb{E}_{s' \sim \pi^\ell} \left[ \log(\sigma(f^{\text{MVR}}(s) - f^{\text{MVR}}(s'))) \right]. \tag{9}$$

This additive form serves as a *state-dependent reward shaping* term rather than a fixed-weight sum of unrelated signals. Since we do not have access to samples of $\pi^\ell$, Eq. (9) is computed using samples from a reference set $\mathcal{D}^{\text{ref}}$ that contains state sequences with top-$k$ video-text similarity scores (aggregated across viewpoints) collected during policy learning. In literature, a similar problem is solved by learning a separate policy solely from VLM-based rewards (Fu et al., 2024), yet such approach fails when $r^{\text{VLM}}$ alone cannot solve the task. Moreover, sampling from $\mathcal{D}^{\text{ref}}$ bears an interesting analogy to human skill learning: trying to repeat our previous good trials is an effective learning strategy for consolidating our understanding of a skill.

**Automatic shaping.** As the policy improves, $r^{\text{MVR}}$ naturally relinquishes control to $r^{\text{task}}$: when the state distribution induced by $\pi$ aligns with the reference set $\mathcal{D}^{\text{ref}}$, the expectation in Eq. (9) approaches zero, causing $r^{\text{VLM}}$ to vanish automatically. This provides strong early guidance that naturally decays as behaviors improve, avoiding persistent conflicts with task objectives.

**The Entire Framework** Fig. 2 illustrates MVR's operation. During policy learning, MVR renders the agent's latest trajectories into videos at a frequency $T^{\text{render}}$. It then queries a VLM to obtain video-text similarity scores and video embeddings, augmenting the dataset $\mathcal{D}$ (lines 10–13). $\mathcal{D}^{\text{ref}}$ is updated at the same time. At a frequency $T^{\text{update}}$, MVR updates $f^{\text{MVR}}$ using Eq. (5) and samples in $\mathcal{D}$ (line 16–18). Importantly, for off-policy algorithms such as TQC (Kuznetsov et al., 2020), the rewards in the agent's replay buffer must be recalculated after $f^{\text{MVR}}$ is updated. The full pseudocode of MVR is provided in Algorithm A1 in the appendix.

# 5 EXPERIMENTS

This section presents the task performance of MVR, followed by an ablation study for its components and results for the influence of using multiple views. We also showcase the importance of combining visual guidance with task rewards using a case study.

## 5.1 SETUP

**Tasks** We evaluate MVR on a total of 19 tasks from two robotics benchmarks: Humanoid-Bench (Sferrazza et al., 2024) and MetaWorld (Yu et al., 2019). Tasks from HumanoidBench are well-suited for exploring the use of VLMs in skill learning, as they encompass static posture generation (stand, balance_simple, sit_simple, balance_hard, and sit_hard) and dynamic motion generation (run, walk, stair, and slide). Meanwhile, tasks from MetaWorld (hammer, push-wall, faucet-close, push-back, stick-pull, handle-press-side, push, shelf-place, window-close, peg-unplug-side) are used to evaluate performance in fine-grained manipulation tasks across diverse visual contexts. See Appxs. A.3 and A.4 for detailed descriptions of the tasks considered in this paper. Among the 27 tasks available in HumanoidBench, we follow the benchmark authors and prior work by focusing on the nine locomotion and posture tasks listed above, excluding loco-manipulation tasks and the most challenging locomotion task *Hurdle*, which do not yield meaningful learning progress within a 10M-step budget. For MetaWorld, we adopt the CW10 subset from Continual World (Wołczyk et al., 2021), which contains ten single-object manipulation tasks that are commonly evaluated under a 1M-step budget.

**Evaluation Metrics** For HumanoidBench tasks, agents are trained for ten million environment steps, and we report the mean and standard deviation of the episodic returns over three random seeds. A higher score indicates better performance. For MetaWorld tasks, agents are trained for one million environment steps, and we report the mean and standard deviation of the task success rates over five random seeds. In our ablation studies on HumanoidBench, to showcase the relative effect of MVR's components, we normalize the performance of MVR's variants using the performance of MVR. In addition, we report the average rank of methods to assess the overall performance for a domain. The lower the better.

**Alternative Methods** For methods only using $r^{\text{task}}$, we select two recent RL algorithms: TQC (Kuznetsov et al., 2020) and DreamerV3 (Hafner et al., 2023). For existing methods that leverage VLMs, we report results for VLM-RM (Rocamonde et al., 2024) and RoboCLIP (Sontakke

Table 1: **MVR outperforms VLM-RM and RoboCLIP for HumanoidBench tasks.** The first four tasks are dynamic tasks and the last five are static tasks. We compare MVR with TQC (Kuznetsov et al., 2020), DreamerV3 (Hafner et al., 2023), VLM-RM (Rocamonde et al., 2024), and RoboCLIP (Sontakke et al., 2024). The DreamerV3 originate from the TD-MPC2 benchmark suite (Hansen et al., 2024). MVR performs the best for five tasks and has the best rank averaged over all tasks. VLM-RM performs the best for two tasks. RoboCLIP cannot perform well for these tasks.[2]

| Task | MVR | TQC | VLM-RM | RoboCLIP | DreamerV3 | Success / Max Return |
|---|---|---|---|---|---|---|
| Walk | $\mathbf{927.47 \pm 1.83}$ ✓ | $510.58 \pm 299.16$ | $535.35 \pm 355.18$ | $737.34 \pm 194.62$ ✓ | $800.2 \pm 158.7$ ✓ | 700 / 1000 |
| Run | $\mathbf{749.23 \pm 56.82}$ ✓ | $647.87 \pm 186.98$ | $14.93 \pm 1.11$ | $501.15 \pm 179.71$ | $633.8 \pm 222.4$ | 700 / 1000 |
| Stair | $208.60 \pm 166.22$ | $\mathbf{282.95 \pm 120.54}$ | $44.96 \pm 4.12$ | $211.33 \pm 56.25$ | $131.1 \pm 43.6$ | 700 / 1000 |
| Slide | $\mathbf{735.03 \pm 142.85}$ ✓ | $514.91 \pm 106.36$ | $163.13 \pm 41.22$ | $494.20 \pm 21.66$ | $436.5 \pm 200.1$ | 700 / 1000 |
| Stand | $\mathbf{918.55 \pm 29.30}$ ✓ | $576.59 \pm 371.0$ | $728.69 \pm 102.19$ | $849.73 \pm 108.86$ ✓ | $622.7 \pm 404.8$ | 800 / 1000 |
| Sit_Simple | $\mathbf{861.07 \pm 19.35}$ ✓ | $822.07 \pm 101.28$ ✓ | $293.72 \pm 65.46$ | $553.90 \pm 244.51$ | $\mathbf{891.4 \pm 38.4}$ ✓ | 750 / 1000 |
| Sit_Hard | $\mathbf{756.67 \pm 108.79}$ ✓ | $511.85 \pm 155.45$ | $322.95 \pm 45.70$ | $559.38 \pm 192.00$ | $433.4 \pm 355.9$ | 750 / 1000 |
| Balance_Simple | $299.87 \pm 58.45$ | $256.79 \pm 54.23$ | $\mathbf{654.42 \pm 335.19}$ | $215.81 \pm 18.19$ | $19.8 \pm 7.0$ | 800 / 1000 |
| Balance_Hard | $95.78 \pm 11.03$ | $92.35 \pm 10.76$ | $\mathbf{107.36 \pm 24.93}$ | $97.03 \pm 18.13$ | $45.9 \pm 27.4$ | 800 / 1000 |
| Average Rank | **1.67** | 3.11 | 3.78 | 2.89 | 3.56 | — |

✓ indicates that the mean return exceeds the official HumanoidBench success threshold.

et al., 2024). VLM-RM combines $r^{\text{task}}$ with image-text similarity scores[1]. RoboCLIP uses video-text similarity but generates a single reward value for each trajectory. Since VLM-RM and RoboCLIP use different RL algorithms in their original implementation, we implement both on top of TQC to ensure fair comparison with MVR. All VLM-based baselines, including RoboCLIP, shape their policies with both the task reward and their respective VLM-derived signals so that differences stem purely from the multimodal guidance design.

Table 2: **MVR achieves higher success rate on Metaworld tasks.** We use the results for DreamerV3 reported by TD-MPC2 benchmark suite (Hansen et al., 2024). The three methods that incorporate visual guidance, MVR, VLM-RM, and RoboCLIP, all outperform TQC and DreamerV3. Within them, MVR achieves the highest success rate, which is followed by RoboCLIP. These results support our claim for the importance of using visual feedback and derive the feedback from multi-view videos.

| Task | MVR | TQC | VLM-RM | RoboCLIP | DreamerV3 |
|---|---|---|---|---|---|
| hammer | $0.20 \pm 0.24$ | $0.20 \pm 0.40$ | $\mathbf{0.40 \pm 0.49}$ | $0.00 \pm 0.00$ | $0.10 \pm 0.17$ |
| push-wall | $\mathbf{0.60 \pm 0.49}$ | $0.10 \pm 0.20$ | $0.40 \pm 0.37$ | $\mathbf{0.60 \pm 0.37}$ | $0.00 \pm 0.00$ |
| faucet-close | $\mathbf{1.00 \pm 0.00}$ | $\mathbf{1.00 \pm 0.00}$ | $\mathbf{1.00 \pm 0.00}$ | $\mathbf{1.00 \pm 0.00}$ | $0.13 \pm 0.15$ |
| push-back | $0.50 \pm 0.45$ | $\mathbf{0.60 \pm 0.37}$ | $0.30 \pm 0.40$ | $0.20 \pm 0.40$ | $0.37 \pm 0.38$ |
| stick-pull | $\mathbf{0.20 \pm 0.24}$ | $0.10 \pm 0.20$ | $0.00 \pm 0.00$ | $0.00 \pm 0.00$ | $0.00 \pm 0.00$ |
| handle-press-side | $\mathbf{1.00 \pm 0.00}$ | $\mathbf{1.00 \pm 0.00}$ | $\mathbf{1.00 \pm 0.00}$ | $\mathbf{1.00 \pm 0.00}$ | $0.77 \pm 0.17$ |
| push | $0.10 \pm 0.20$ | $0.10 \pm 0.20$ | $0.00 \pm 0.00$ | $\mathbf{0.30 \pm 0.40}$ | $0.00 \pm 0.00$ |
| shelf-place | $\mathbf{0.20 \pm 0.24}$ | $0.00 \pm 0.00$ | $0.00 \pm 0.00$ | $\mathbf{0.20 \pm 0.25}$ | $0.00 \pm 0.00$ |
| window-close | $\mathbf{1.00 \pm 0.00}$ | $\mathbf{1.00 \pm 0.00}$ | $\mathbf{1.00 \pm 0.00}$ | $\mathbf{1.00 \pm 0.00}$ | $0.61 \pm 0.16$ |
| peg-unplug-side | $0.60 \pm 0.37$ | $0.30 \pm 0.40$ | $0.60 \pm 0.49$ | $\mathbf{0.80 \pm 0.40}$ | $0.77 \pm 0.19$ |
| Average Rank | **1.50** | 2.20 | 2.40 | 2.00 | 3.90 |

**Ablations** We compare MVR with several variants to validate its components. The variant *w/o reg* removes $L_{\text{reg}}$ from Eq. (5). For *w/o reference*, we use the raw relevance $f^{\text{MVR}}(s)$ rather than Eq. (9) as visual guidance, ablating the state-dependent expectation matching (and its natural decay). The variant *MVR-CLIP* uses image-text similarity computed by a CLIP model. For the variant *direct*, we train $f^{\text{VLM}}$ by directly fitting it to the video-text similarity scores. Moreover, we compare the results using one to four views to analyze the influence of using multiple viewpoints. Lastly, we compare the results of using ViCLIP-L (428M parameters) and ViCLIP-B (150M parameters) to study the influence of VLM size. Additional ablations on reward components and alternative temporal pooling strategies are summarised in Tabs. A10 and A11.

**Implementation Details** MVR renders one trajectory out of every nine into a video ($T^{\text{render}} = 9$). For each rendered episode we draw one viewpoint uniformly at random; across training this rotates

---

[1]While Rocamonde et al. (2024) only uses VLM-based rewards for pose generation, it is difficult to solve locomotion tasks without $r^{\text{task}}$.

[2]Success thresholds follow the official HumanoidBench specification (Sferrazza et al., 2024); the maximum returns correspond to the theoretical upper bounds defined in the benchmark.

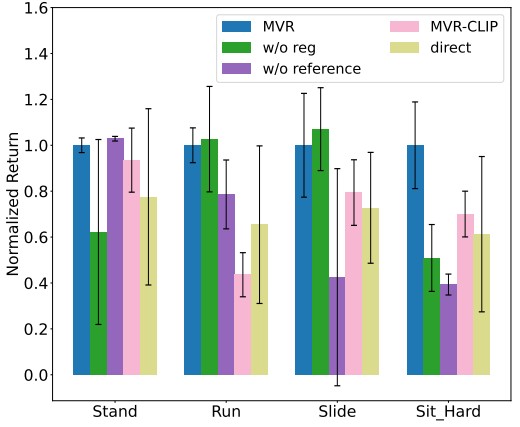
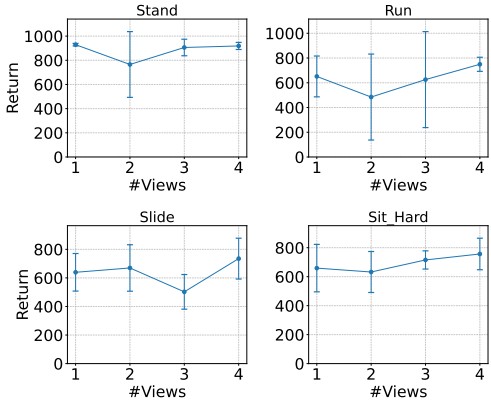

(a) **The use of videos, Eq. (3), Eq. (4), and Eq. (9) are all useful.** The variant *w/o reg* and *w/o reference* stand for disabling Eq. (4) and Eq. (9). *MVR-CLIP* and *direct* means using images and fitting the $f^{\mathrm{MVR}}$ to the similarity scores rather than using Eq. (3). No variant has consistently good performance as MVR does.

(b) **Using more views is beneficial.** These figures show the performance of MVR when using one to four views. For Stand, one view seems sufficient as the return is close to optimal. For other tasks, a general trend is that performance improves when using more views, though fluctuation exists.

Figure 3: Method ablation and the influence of the number of views.

through all cameras without exceeding the rendering/encoding budget of a single-view pipeline. For each video, MVR extracts segments of length 64 ($T^{\mathrm{video}} = 64$) and queries the ViCLIP model (Wang et al., 2024a) for video-text similarity scores. The relevance model is updated every 100,000 environment steps ($T^{\mathrm{update}} = 100,000$) with early stopping. The reference set $\mathcal{D}^{\mathrm{ref}}$ keeps the top-$k$ most relevant state sequences observed so far with $k = 10$ for all experiments, i.e., only the highest-quality trajectories remain in memory. The hyperparameter $w$ in Eq. (6) is selected from $\{0.01, 0.1, 0.5\}$ via grid-search. In contrast, VLM-RM queries VLMs at each policy learning step, which is not scalable for the tasks considered here. Therefore, we adopt the same rendering frequency as MVR and fit a reward model for policy learning. Both MVR and RoboCLIP use ViCLIP-L (428M parameters). VLM-RM uses CLIP-ViT-H-14 (986M parameters). More details can be found in Appx. A.7.

Table 3: **MVR can aggregate information from multiple views.** This table compares the task performance of using four different viewpoints only and random viewpoints. While the best viewpoint is task-dependent, using random viewpoints (MVR) has the best performance for Slide and Sit_Hard and second-best performance for Run, indicating that MVR can aggregate multi-view information.

| Task | 0° | 90° | 180° | 270° | MVR |
|---|---|---|---|---|---|
| Stand | $914.17 \pm 32.54$ | $929.12 \pm 10.58$ | $912.20 \pm 26.66$ | $\mathbf{931.10 \pm 32.08}$ | $918.55 \pm 29.30$ |
| Run | $611.43 \pm 294.44$ | $651.23 \pm 164.96$ | $\mathbf{875.33 \pm 6.44}$ | $567.91 \pm 159.47$ | $749.23 \pm 56.82$ |
| Slide | $559.59 \pm 0.00$ | $639.28 \pm 131.48$ | $647.18 \pm 154.73$ | $657.64 \pm 205.81$ | $\mathbf{735.03 \pm 142.85}$ |
| Sit_Hard | $528.14 \pm 176.60$ | $659.10 \pm 163.77$ | $452.15 \pm 210.89$ | $723.72 \pm 112.11$ | $\mathbf{756.67 \pm 108.79}$ |

## 5.2 RESULTS

**Proficiency for Soliciting Visual Guidance** Tab. 1 presents results for the task performance of HumanoidBench tasks. MVR has the best performance for five tasks. VLM-RM outperforms other methods for two tasks, but RoboCLIP cannot outperform others for any tasks. Notably, MVR has the best average rank for these tasks, corroborating its proficiency for soliciting visual guidance. The comparison among MVR, VLM-RM, and TQC reveals an interesting insight. VLM-RM performs worse than TQC for three dynamic tasks but surpasses it for three static tasks. In contrast, MVR outperforms TQC for three dynamic tasks and all static tasks. Thus, generating visual guidance from videos not only applies to challenging dynamic tasks but is also more effective for static tasks.

Tab. 2 shows the results for MetaWorld tasks. MVR attains the highest average success rate and has the best average rank, followed by RoboCLIP. Despite RoboCLIP's strong overall performance

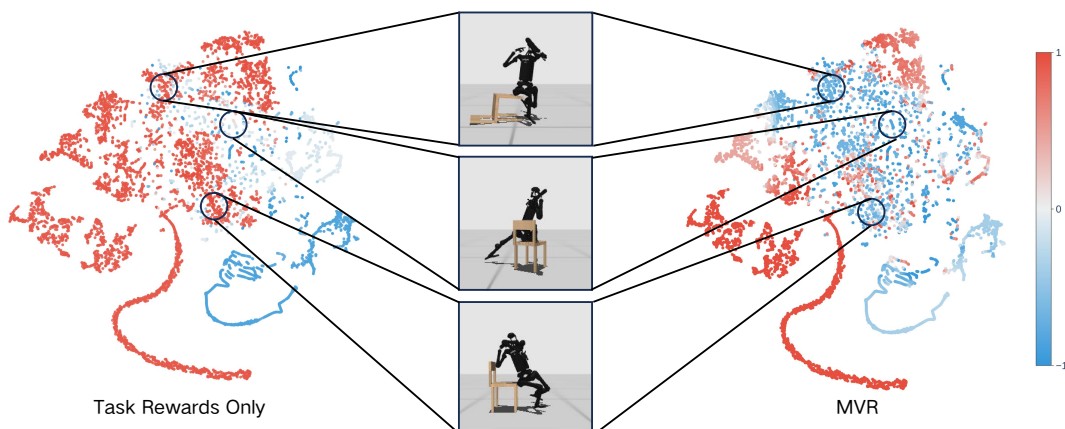

Figure 4: **MVR identifies suboptimal states through reward shaping.** We visualize states generated by a TQC agent for the Sit_Hard task and annotate them with $r^{\text{task}}$ (left) and the shaped reward $r^{\text{MVR}} = r^{\text{task}} + w r^{\text{VLM}}$ (right) computed by a trained MVR agent. While task rewards are high when the agent is close to the chair, MVR's visual guidance component assigns low values to improper sitting poses (sitting on chair's leg, leaning, or sitting at the edge), effectively shaping the reward landscape to discourage these visually suboptimal but task-rewarding states.

(second in average success rate and rank), it scored zero on two specific tasks, indicating potential instability. VLM-RM has a higher average success rate than TQC but a worse average rank, indicating inferior performance on certain tasks.

The performance margins between multi-view methods are smaller on MetaWorld than on Humanoid-Bench for two reasons. First, MetaWorld episodes are short (500 steps) and focus on single tabletop objects, where sparse trajectory-level feedback already provides useful exploration hints. Second, the long-horizon HumanoidBench locomotion suite demands dense state-conditioned guidance across multiple, occasionally occluded joints, which explains the larger performance gaps observed in Tab. 1.

Meanwhile, readers may notice that, among the 19 tasks considered in this paper, MVR performs worse than TQC for four tasks, indicating that using $r^{\text{MVR}}$ might compromise the optimization of $r^{\text{task}}$. A possible reason is that the underlying VLM is trained on mostly human activities rather than robotics data, and we leave the few-shot extension of MVR for future work. **Model Ablation** We now discuss the results presented in Fig. 3a from three perspectives. Firstly, the inferior performance of *direct* and the unstable performance of *w/o reg* indicate that both Eq. (3) and Eq. (4) are essential for learning the relevance model. Moreover, the decreased performance of MVR-CLIP shows that assessing agent behaviors with videos is critical for deriving visual guidance, further confirming the insight discussed above. Lastly, the variant *w/o reference* is outperformed by MVR for three tasks, confirming the efficacy of our proposed reward function Eq. (9). Altogether, these results demonstrate the effectiveness of components in MVR.

Table 4: **MVR with small and large VLMs.**

| Task | ViCLIP-L (428M) | ViCLIP-B (150M) |
|---|---|---|
| Stand | $\mathbf{918.55 \pm 29.30}$ | $917.97 \pm 30.71$ |
| Run | $\mathbf{749.23 \pm 56.82}$ | $517.25 \pm 383.12$ |
| Slide | $735.03 \pm 142.85$ | $\mathbf{778.84 \pm 24.80}$ |
| Sit_Hard | $\mathbf{756.67 \pm 108.79}$ | $689.50 \pm 234.37$ |

As shown in Table 4, a comparison between the ViCLIP-B and ViCLIP-L models reveals that the larger ViCLIP-L model yields better performance in three of the four evaluated tasks, confirming the common hypothesis of the scaling law. In addition, we present the results for using different RL algorithms in MVR in Appx. B.1 and the results for using different values for the weight $w$ of $r^{\text{VLM}}$ in Appx. B.8.

**The Influence of Views** Fig. 3b presents the results of MVR when using different numbers of views. While using one view is enough for the Stand task, it is beneficial to increase the number of views for other tasks, which confirms our claim for using multi-view videos for computing $r^{\text{VLM}}$. Meanwhile, Tab. 3 shows the results of different views and compares them with MVR, which selects views uniformly at random. The best choice of view is task-dependent, which can be explained by the difference in the required motion patterns. For example, for Sit_Hard, the two side views 90° and 270° are more effective than the rest two because it is easier to determine the robot-chair distance

from side views. Notably, moving from one view to four orthogonal views raises the mean returns on Run, Slide, and Sit_Hard by +98, +96, and +98 respectively, pushing them past the HumanoidBench success thresholds (700, 700, 750) and turning previously failing policies into successful ones. Since the HumanoidBench returns are capped at 1000, these gains amount to roughly 10% of the total attainable return. Interestingly, we can obtain competitive or even better results by randomly sampling viewpoints, which highlights MVR's capability in aggregating information in multi-view videos.

**Case Study** Finally, we showcase the significance of visual guidance using the Sit_Hard. Fig. 4 is a t-SNE (van der Maaten & Hinton, 2008) visualization of 20 episodes produced by a trained TQC agent. We annotate the states with $r^{\text{task}}$ and $r^{\text{MVR}}$ computed by a trained MVR agent in Fig. 4. The rewards are normalized to the range [-1, 1] for visualization purpose.

A general trend in Fig. 4 is that $r^{\text{task}}$ and $r^{\text{MVR}}$ align in many parts of the state space. For example, both reward functions assign high rewards to the bottom left part of the space and low rewards to the bottom right. However, upon closer inspection of the regions where the two reward functions diverge, we gain valuable insights. Specifically, the center region corresponds to undesirable sitting poses, such as sitting on the chair's leg (the top example), leaning on the chair (the middle example), and sitting at the chair's edge (the bottom example). Although these poses receive high values of $r^{\text{task}}$ due to the minimal distance between the robot and the chair, they are inherently unstable and suboptimal. Notably, MVR is able to identify them, which explains its superior performance over TQC in this task. Without visual guidance, a much larger number of online samples will be required for recognizing such *visually* suboptimal behaviors by their long-term consequences. This emphasizes the importance of visual guidance in learning general and versatile skills.

*Quantitative analyses.* Appendix B.7 details the correlation between the learned shaping signals and binary success, and also documents the temporal decay of $r^{\text{VLM}}$ that drives the automatic handover to $r^{\text{task}}$.

## 6 CONCLUSIONS

In this paper, we present a novel framework named Multi-view Video Reward to generate visual guidance using VLMs for RL agents. In contrast to existing attempts that use VLMs to generate rewards, MVR leverages videos collected from multiple viewpoints to evaluate agent behaviors, which enables it to characterize dynamic motions and overcome occlusions. Moreover, we propose a state-dependent reward shaping formulation that integrates task rewards with VLM-based guidance and automatically reduces the influence of the auxiliary signal as the behavior aligns with the desired motion pattern. We evaluate MVR for nine humanoid tasks and ten manipulation tasks. MVR has the best average rank when compared to existing methods that use VLMs and methods that only use task rewards, confirming its superiority for soliciting visual guidance for RL agents. Furthermore, we confirm the design of MVR with an ablation study and investigate the influence of using multiple views. Lastly, we present a case study to showcase the importance of visual feedback for fostering the desired motion of a task.

**Ethics Statement.** This work uses only simulated benchmarks, does not collect human data, and does not interact with sensitive or safety-critical systems. To the best of our knowledge, it poses no foreseeable ethical, privacy, or security risks.

**Reproducibility Statement.** Section 4.1 and Alg. A1 specify the training objectives, optimization steps, and update schedule for MVR. Implementation details covering architectures, environment settings, rendering schedules, and all hyperparameters appear in Appendix A.1, while additional ablations and alternative backbones are summarized in Appendices B.1 and B.6. Appendix B compiles the evaluation tables cited in the main text. The submission bundles the Hydra configuration files, evaluation scripts, and fixed random seeds required to reproduce every experiment.

## ACKNOWLEDGMENTS

C. Fang was supported by the National Science and Technology Major Project (No. 2022ZD0114902).

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

---

**Algorithm A1** The Pseudocode of MVR

---

1: **Input:** textual goal description $\ell$, environment steps $T^{\text{env}}$, video sampling frequency $T^{\text{render}}$, model update frequency $T^{\text{update}}$
2: Initialize the policy $\pi$ and relevance model $f^{\text{MVR}}$
3: Initialize the replay buffer of MVR: $\mathcal{D} = \emptyset$.
4: Initialize the reference set: $\mathcal{D}^{\text{ref}} = \emptyset$.
5: **for** t = 0, 2, ..., $T^{\text{env}}$ **do**
6:     Observe state $s_t$, apply action $a_t \sim \pi(\cdot|s_t)$ and observe the next state $s_{t+1}$ and reward $r_t^{\text{task}}$.
7:     Compute $r_t^{\text{VLM}}$ and form the shaped reward $r_t^{\text{MVR}} = r_t^{\text{task}} + w r_t^{\text{VLM}}$.
8:     Use $s_t, a_t, s_{t+1}$, and $r_t^{\text{MVR}}$ to update $\pi$.
9:     **if** t > 0 **then**
10:         **if** t % $T^{\text{render}}$ = 0 **then**
11:             Render the last episode into videos.
12:             Augment $\mathcal{D}$ with the latest state sequences, videos, and their video-text similarity scores:
                $\mathcal{D} = \mathcal{D} \cup \{(\mathbf{s}, \mathbf{o}, \psi^{\text{VLM}}(\mathbf{o}, \ell))\}$.
13:             Switch to the next viewpoint.
14:             Update the reference set $\mathcal{D}^{\text{ref}}$ if necessary.
15:         **end if**
16:         **if** t % $T^{\text{update}}$ = 0 **then**
17:             Use data in $\mathcal{D}$ to update $f^{\text{MVR}}$ with Eq. (5).
18:         **end if**
19:     **end if**
20: **end for**

---

# A  EXPERIMENTAL SETUP

## A.1  IMPLEMENTATION DETAILS

In this study, we adopted the JAX implementation of TQC from the SBX library (Raffin et al., 2021) as our baseline RL algorithm. The parameters of this algorithm were subsequently fine-tuned to address the complexities inherent in humanoid robot control tasks. Our proposed Multi-view Video Rewards (MVR) framework builds upon this TQC foundation. The experiments were conducted using the most recent version of the HumanoidBench simulation environment (Sferrazza et al., 2024).

All computational experiments were performed on NVIDIA RTX 4090 GPUs. Under this hardware configuration, each MetaWorld task required approximately 25 minutes to complete, while each HumanoidBench task took roughly 4 hours. The specific versions of key software packages utilized throughout our experiments are enumerated below:

- Python 3.11
- Torch 2.3.1
- TorchVision 0.18.1
- JAX 0.4.37
- JAXLib 0.4.36
- Flax 0.10.2
- Gymnasium 0.29.1
- SBX-RL 0.18.0

## A.2  MVR ALGORITHM PSEUDOCODE

For completeness, we present the full pseudocode of the proposed MVR framework below; see Fig. 2 in the main text for the high-level pipeline.

## A.3  HUMANOIDBENCH

Humanoid robots offer significant promise for assisting humans in a multitude of environments and tasks, primarily due to their human-like morphology, which grants them flexibility and adaptability. HumanoidBench (Sferrazza et al., 2024) is a significant contribution in this area, offering a simulated benchmark with 27 distinct whole-body control tasks. These tasks are designed to present unique and substantial challenges, including those requiring intricate long-horizon control and sophisticated

coordination. As demonstrated by (Sferrazza et al., 2024), even state-of-the-art RL algorithms face difficulties with many of these tasks. For our study, we excluded tasks that were identified as poorly designed (e.g., the 'powerlift' task lacked appropriate reward signals for lifting the barbell), resulting in a set of nine valid tasks for evaluation.

A brief description of each selected HumanoidBench task is provided below.

**Stand**   The robot's objective is to maintain an upright standing posture for the duration of the trial. This requires dynamic adjustments to ensure stability, compensating for environmental changes or internal balance shifts. This task evaluates the robot's capacity for stable positioning, a fundamental skill for stationary operations and transitions between movements.

**Walk**   This task involves the robot maintaining a consistent forward velocity of approximately 1 m/s along the global x-axis, without falling. It challenges the robot's balance, coordination, and its ability to generate efficient forward locomotion, crucial for navigation in varied settings.

**Run**   Here, the robot must achieve and maintain a forward running speed of 5 m/s. This higher-speed task demands advanced coordination, strength, and precise control over leg movements. It involves not just maintaining velocity but also managing the complex dynamics of transitioning between walking and running gaits.

**Stair**   The robot is tasked with traversing a repeating sequence of upward and downward stairs at a steady speed of 1 m/s. This requires the robot to adapt its gait and step height for each stair while preserving balance and forward momentum, simulating real-world obstacle navigation.

**Slide**   In this task, the robot navigates an alternating sequence of upward and downward inclines (slides) at a speed of 1 m/s. The primary challenge lies in adapting to varying surface inclinations and maintaining stability on potentially slippery or uneven terrain, testing real-time gait adjustment capabilities.

**Sit_Simple**   The robot must sit on a chair positioned closely behind it. This task tests the robot's environmental perception, motion planning for the sitting sequence, and controlled execution of the sit-down action while maintaining stability.

**Sit_Hard**   This task increases the complexity of sitting by randomizing the chair's initial position. Furthermore, the chair can move due to forces exerted by the robot, meaning its position and orientation can change dynamically. The robot must continuously adapt its approach, handling the uncertainty of the chair's movement while maintaining balance.

**Balance_Simple**   The robot is required to maintain balance on an unstable board that pivots on a fixed spherical point. The core challenge is to remain stable on a surface that can tilt in any direction, demanding continuous postural adjustments and center of mass control.

**Balance_Hard**   This task extends the balancing challenge by introducing a moving pivot point for the unstable board. This adds a layer of complexity, as the robot must simultaneously adjust for the board's instability and the dynamic movement of its pivot, necessitating a high degree of coordination and real-time responsiveness.

Prompts derived from these task descriptions are summarized in Tab. A1.

## A.4 METAWORLD

Meta-World is an open-source simulated benchmark for meta-RL and multi-task learning consisting of 50 distinct robotic manipulation tasks. We selected 10 Metaworld tasks in CW10 (Wołczyk et al., 2021) because their difficulty is suitable for solving in 1M steps. A brief introduction to each selected task is summarized below.

Table A1: Text instructions for humanoid robot tasks.

| Prompt | Description |
|---|---|
| Stand | a humanoid robot standing |
| Walk | a humanoid robot walking |
| Run | a humanoid robot running |
| Stair | a humanoid robot walking |
| Slide | a humanoid robot walking |
| Sit_Simple | a humanoid robot sitting on a chair |
| Sit_Hard | a humanoid robot sitting on a chair |
| Balance_Simple | a humanoid robot balancing on the board |
| Balance_Hard | a humanoid robot balancing on the board |

**Hammer**   In this task, the robot gripper is required to pick up a hammer and use it to strike a nail or a similar target on a surface. This task tests the robot's abilities in grasping, tool manipulation, force control, and aiming accuracy.

**Push-Wall**   This task requires the robot gripper to push an object, such as a puck or cylinder, towards a target location. However, a wall obstructs a direct path, so the robot must navigate the object around or over the wall. This task challenges path planning, obstacle avoidance, and pushing mechanics.

**Faucet-Close**   The robot gripper must turn a faucet handle clockwise to close it. This involves precise grasping of the handle and applying the correct rotational force. It tests fine motor control and the understanding of object affordances.

**Push-Back**   In this task, the robot gripper is required to push a wooden block backward to a designated goal position. This primarily tests the robot's ability to apply controlled force in a specific direction and manage the displacement of the object.

**Stick-Pull**   The robot gripper needs to grasp a stick and pull it towards itself or out of a fixture. This task assesses grasping strength, pulling force control, and potentially the ability to dislodge an object from its holder.

**Handle-Press-Side**   This task requires the robot gripper to press a handle downwards, approaching it from the side. This tests the robot's ability to apply force at a specific angle and direction, often simulating actions like opening a door or activating a mechanism.

**Push**   This is a fundamental pushing task where the robot gripper must push an object, typically a cylinder or puck, to a target location on a flat surface. It evaluates the robot's basic pushing capabilities and goal-oriented movement.

**Shelf-Place**   The robot gripper is tasked with picking up an object, such as a cube, and then accurately placing it onto a designated spot on a shelf. This involves a sequence of grasping, lifting, 3D path planning, and precise placement.

**Window-Close**   In this task, the robot gripper needs to close a window. This might involve pushing a sliding window or manipulating a latch or handle, depending on the window type. The task tests interaction with larger articulated objects and achieving a specific state change in the environment.

**Peg-Unplug-Side**   The robot gripper must grasp a peg that is inserted into a hole and then unplug it by pulling it out sideways. This requires precise grasping, applying force in a specific lateral direction, and successfully disengaging the parts.

Prompts derived from these task descriptions are summarized in Table A2.

Table A2: Text instructions for MetaWorld robot tasks.

| Prompt | Description |
|---|---|
| hammer | a robot gripper hammering a screw on the wall |
| push-wall | a robot gripper bypassing a wall and pushing a cylinder |
| faucet-close | a robot gripper rotating the faucet clockwise |
| push-back | a robot gripper pushing the wooden block backward |
| stick-pull | a robot gripper pulling a stick |
| handle-press-side | a robot gripper pressing a handle down sideways |
| push | a robot gripper pushing a cylinder |
| shelf-place | a robot gripper picking and placing a cube onto a shelf |
| window-close | a robot gripper pushing and closing a window |
| peg-unplug-side | a robot gripper unplugging a peg sideways |

## A.5 ARCHITECTURE AND HYPERPARAMETERS

**RL Training** Each humanoid task was trained for 10 million steps, and each metaworld task was trained for 1 million, following the setup outlined in (Sferrazza et al., 2024; Hansen et al., 2024). Samples were collected from 8 parallel environments, with the learning rate decaying from $6 \times 10^{-4}$ at the start to $5 \times 10^{-5}$ at the final step, ensuring training stability. The policy was updated by sampling from the replay buffer with a batch size of 256. Every 16 steps, MVR performs 16 gradient updates. Both actor and critic networks consist of three-layer neural networks, each with a width of 256 and a quantile count of 50, while the number of top quantiles to drop per network is set to 5. SDE (Raffin et al., 2022) was enabled to enhance the algorithm's exploration. The key parameters are summarized in Tab. A3.

**Compute cost** For each rendered trajectory we sample only one viewpoint, so the wall-clock cost matches that of a single-view implementation. A HumanoidBench run takes roughly four hours on one NVIDIA RTX 4090 using MuJoCo's EGL backend (about 20 GB GPU memory). When GPU rendering is unavailable we fall back to the OSMesa backend, which requires approximately 2.5 GB of CPU memory. MetaWorld tasks are shorter; each run finishes in about 25 minutes and needs roughly 2 GB of GPU memory. These numbers are consistent with the settings reported in the official HumanoidBench repository.

**Reward Model** The proposed reward model is composed of two core components: a multi-layer backbone network and a parameterized predictor vector. As outlined in Tab. A3, the backbone adopts a fully-connected architecture with two hidden layers of dimension 512, activated by ReLU nonlinearity. The predictor is implemented as a single learnable vector of dimension 512, initialized via Kaiming uniform distribution to stabilize gradient dynamics during training. Formally, given an input state $\mathbf{s} \in \mathbb{R}^d$, the backbone first projects it into a latent embedding space through sequential transformations:

$$\mathbf{h}_1 = \text{ReLU}(\mathbf{W}_1\mathbf{s} + \mathbf{b}_1), \quad \mathbf{h}_2 = \mathbf{W}_2\mathbf{h}_1 + \mathbf{b}_2 \tag{A1}$$

where $\mathbf{W}_1 \in \mathbb{R}^{512 \times d}$, $\mathbf{W}_2 \in \mathbb{R}^{512 \times 512}$ are weight matrices. The final embedding $\mathbf{e} = \mathbf{h}_2/\|\mathbf{h}_2\|_2$ is $\ell_2$-normalized along the feature dimension to constrain the output scale. The predictor $\mathbf{p} \in \mathbb{R}^{512}$ operates as a direction-sensitive projection head. It is normalized as $\tilde{\mathbf{p}} = \mathbf{p}/\|\mathbf{p}\|_2$ and broadcasted across the batch dimension. The value $f^{\text{MVR}}$ is computed via cosine similarity between the normalized embedding and the predictor:

$$f^{\text{MVR}} = \langle \mathbf{e}, \tilde{\mathbf{p}} \rangle = \sum_{i=1}^{512} e_i \cdot \tilde{p}_i \tag{A2}$$

This design bounds reward outputs within $[-1, 1]$, promoting training stability. Optionally, the model can return the intermediate embedding $\mathbf{e}$ for auxiliary tasks or interpretability analysis.

**Reward Model Learning** MVR renders one out of every nine trajectories into videos with a resolution of $224 \times 224$. From each video, MVR extracts segments of length 64 and queries the ViCLIP model (Wang et al., 2024a) for video-text similarity scores, using a batch size of 32. These

Table A3: Hyperparameters of MVR

| Category | Hyperparameter | Value |
|---|---|---|
| **TQC Network Architecture** | | |
| | Policy Layers | [256, 256, 256] |
| | Critic Layers | [256, 256, 256] |
| | Quantile Count | 50 |
| **Reward Model** | | |
| | Backbone | [512,512] |
| | Predictor | Parameter(512) |
| **RL Training** | | |
| | Total Timesteps $T^{\text{env}}_{humanoidbench}$ | $1 \times 10^7$ |
| | Total Timesteps $T^{\text{env}}_{metaworld}$ | $1 \times 10^6$ |
| | Number of Environments | 8 |
| | Learning Rate | $\text{lin\_}6 \times 10^{-4} \to 5 \times 10^{-5}$ |
| | Batch Size | 256 |
| | Buffer Size | $1 \times 10^6$ |
| | Gradient Steps | 16 |
| | Tau $\tau$ | 0.01 |
| | Gamma $\gamma$ | 0.99 |
| | Sde Sample Frequency | 4 |
| | Top Quantiles to Drop Per Net | 5 |
| **Reward Model Learning** | | |
| | Model | ViCLIP |
| | Image Resolution | $224 \times 224$ |
| | Reward Scale $w$ | $\{0.01, 0.1, 0.5\}_{\text{best}}$ |
| | Video Clip Length $T^{\text{video}}$ | 64 |
| | Relabel Interval | $1 \times 10^5$ |
| | Update Interval $T^{\text{update}}$ | $1 \times 10^5$ |
| | Video Sampling Frequency $T^{\text{render}}$ | 9 |
| | Encoding Batch Size | 32 |
| | Reward Dataset Size | $2 \times 10^4$ |

scores are then added to the reward dataset, which has a size of $2 \times 10^4$. The relevance model is updated every 100,000 environment steps, with early stopping employed to prevent overfitting. The hyperparameter $w$ in Eq. (6) is selected from $\{0.01, 0.1, 0.5\}$ using grid search. MVR relabels the dataset every $1 \times 10^5$ steps. Key parameters are summarised in Tab. A3.

### A.6 TOP-/BOTTOM-RANKED FRAMES IN SIT_HARD

To further clarify where MVR actively modifies the reward landscape, we visualize frames from the Sit_Hard task ranked by the task reward $r^{\text{task}}$ and by the full shaped reward $r^{\text{MVR}} = r^{\text{task}} + wr^{\text{VLM}}$. Figure A1 shows the top- and bottom-ranked frames under both criteria. The top-ranked frames for $r^{\text{task}}$ and $r^{\text{MVR}}$ consistently correspond to clearly successful behaviors in which the humanoid sits stably on the chair with an upright torso, indicating that the visual shaping term preserves the optimal behavior selected by the task reward. The bottom-ranked frames for both rewards are dominated by catastrophic failures, where the robot has fallen off the chair or never approaches a seated configuration, confirming that both signals assign low value to clearly unsuccessful states even though the exact ordering among these failures is less critical for control. This qualitative alignment on clearly successful behaviors is consistent with our goal that $r^{\text{VLM}}$ automatically decays once the policy reliably reaches the desired motion pattern.

In contrast, differences between $r^{\text{task}}$ and $r^{\text{MVR}}$ primarily emerge in the intermediate-reward regime. As illustrated by the analysis in Fig. 4, clusters of states that receive relatively high $r^{\text{task}}$ but correspond to visually unstable sitting poses (e.g., sitting on a chair leg, slipping off the edge, or leaning heavily)

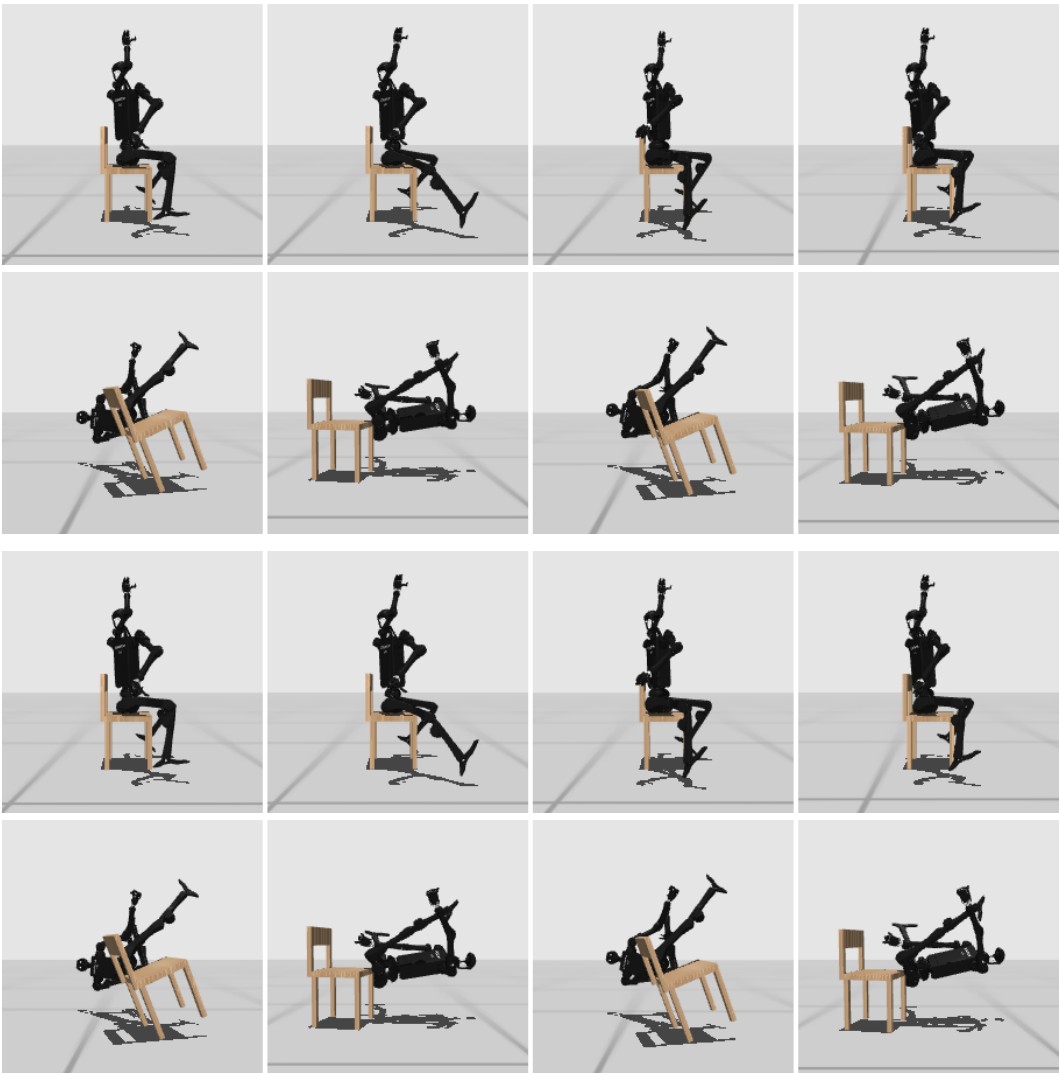

Figure A1: **Top-/bottom-ranked frames by $r^{\text{task}}$ and $r^{\text{MVR}}$ in the Sit_Hard task.** The first two rows show the top and bottom frames under the task reward $r^{\text{task}}$, while the last two rows show the corresponding frames under the shaped reward $r^{\text{MVR}}$. For both rewards, the highest-ranked frames coincide with visually unambiguous successful sitting poses, confirming that MVR preserves the optimal behavior emphasized by the task reward. The bottom-ranked frames are dominated by clearly failed attempts under both signals, serving mainly as a sanity check that obviously bad states receive low values. Together with the qualitative analysis in Fig. 4, this suggests that the main effect of MVR is to reshuffle intermediate-reward states, down-weighting visually unstable but task-rewarding poses while leaving the optimal behavior unchanged.

are systematically down-weighted by $r^{\text{MVR}}$. These mid-range corrections are precisely where MVR is active: the shaped reward preserves the ordering of clearly successful states while sharpening the ranking among borderline cases, assigning lower values to visually unstable yet task-rewarding poses and relatively higher values to visually robust configurations.

### A.7    VLM Baselines

This section provides a detailed overview of the VLM baselines. Tab. A4 summarizes the models used.[1]

Table A4: Models Used in MVR, VLM-RM, and RoboCLIP

| Algorithm | Model |
|-----------|-------|
| **MVR** | ViCLIP |
| **VLM-RM** | CLIP-ViT-H-14-laion2B-s32B-b79K |
| **RoboCLIP** | ViCLIP |

**VLM-RM**    VLM-RM uses a pre-trained vision-language model (VLM) as a zero-shot reward model (RM) to represent tasks in natural language. The original implementation employs CLIP to train a MuJoCo humanoid robot for static tasks, such as kneeling, doing splits, and sitting cross-legged. In this paper, we use CLIP-ViT-H-14-laion2B-s32B-b79K to match the model size with ViCLIP. We select TQC instead of SAC because it demonstrates superior performance and serves as the RL backbone for MVR, eliminating performance discrepancies related to algorithm selection. We directly use MSE loss to fit the CLIP output, avoiding ranking loss to ensure maximum consistency with the original VLM-RM implementation. $r^{\text{VLM-RM}}$ is defined as follows:

$$r^{\text{VLM-RM}}(s) \triangleq \mathbb{E}_{s' \sim \pi^\ell} \left[ f^{\text{VLM-RM}}(s) \right]. \tag{A3}$$

where $f^{\text{VLM-RM}}(s)$ represents the reference function. The remaining parameters of VLM-RM align with MVR (see Tab. A3).

**RoboCLIP**    RoboCLIP generates rewards for online RL agents using pre-trained video-language models. It provides a sparse, end-of-trajectory reward based on the similarity between the agent's video and a target text description. The original version embedded inputs using S3D (Xie et al., 2018) (pre-trained on HowTo100M (Miech et al., 2019)) and used the unscaled scalar product of embeddings as the reward. In this paper, we adapt RoboCLIP by substituting ViCLIP for S3D and replacing the SAC algorithm with TQC to align with the MVR baseline for comparison. The algorithms and corresponding VLM models are summarized in Tab. A4.

## B    Additional Results

This section presents further experimental results to provide a more comprehensive evaluation of MVR and its components.

### B.1    Compatibility with Different RL Algorithms

To evaluate the general applicability of the MVR framework, we integrated it with a different RL algorithm, SAC, in addition to the TQC algorithm used in the main experiments. Tab. A5 compares the performance of MVR-SAC against the baseline SAC algorithm, alongside the MVR-TQC and TQC results for reference. We further pair MVR with Simba (Lee et al., 2025), a large-scale off-policy learner, to confirm that the shaping term scales to stronger architectures.

The results indicate that the performance benefit from VLM-based rewards provided by MVR is less pronounced when using SAC compared to TQC. By contrast, integrating MVR with Simba preserves the gains on the hardest locomotion tasks, demonstrating that multi-view shaping transfers to the latest high-capacity agents (Tab. A6).

---

[1]For fairness, we exclude methods such as RL-VLM-F (Wang et al., 2024b) that require repeated queries to large proprietary VLMs, which falls outside our automated large-scale evaluation setting.

Table A5: **MVR yields greater performance improvements when integrated with the TQC algorithm compared to SAC on complex humanoid tasks.** This table presents average returns for MVR-SAC and MVR-TQC against their SAC and TQC baselines across selected HumanoidBench tasks.

| Task | MVR-SAC | SAC | MVR-TQC | TQC |
|------|---------|-----|---------|-----|
| Stand | $704.24 \pm 329.17$ | $581.77 \pm 371.68$ | $\mathbf{918.55 \pm 29.30}$ | $576.59 \pm 371.0$ |
| Run | $280.24 \pm 103.91$ | $299.39 \pm 314.65$ | $\mathbf{749.23 \pm 56.82}$ | $647.87 \pm 186.98$ |
| Slide | $219.79 \pm 176.13$ | $94.43 \pm 52.89$ | $\mathbf{735.03 \pm 142.85}$ | $514.91 \pm 106.36$ |
| Sit_Hard | $172.46 \pm 124.35$ | $263.87 \pm 97.12$ | $\mathbf{756.67 \pm 108.79}$ | $511.85 \pm 155.45$ |

Table A6: **HumanoidBench returns for Simba with and without MVR shaping.** Simba numbers are taken from the original publication; Simba+MVR averages three seeds under the same horizon.

| Task | Simba | Simba+MVR |
|------|-------|-----------|
| Stand | **906.94** | 814.53 |
| Run | 741.16 | **817.69** |
| Slide | 577.87 | **814.53** |
| Sit_Hard | 783.95 | **785.23** |

## B.2 VLM BACKBONE ABLATION

To assess the sensitivity of MVR to the choice of vision-language backbone, we additionally train an S3D-based variant of MVR on four HumanoidBench tasks and compare it with the ViCLIP-based version and the TQC baseline. Tab. A7 shows that MVR-S3D consistently outperforms TQC and achieves performance comparable to MVR-ViCLIP (slightly better on Run/Stand and slightly worse on Slide/Sit_Hard), suggesting that our gains do not rely on a particular VLM.

Table A7: **Effect of VLM backbone on MVR performance.** Returns on four HumanoidBench tasks for TQC, MVR with ViCLIP, and MVR with S3D.

| Task | TQC | MVR-ViCLIP | MVR-S3D |
|------|-----|------------|---------|
| Run | $647.87 \pm 186.98$ | $749.23 \pm 56.82$ | $\mathbf{793.61 \pm 57.27}$ |
| Slide | $514.91 \pm 106.36$ | $\mathbf{735.03 \pm 142.85}$ | $696.51 \pm 32.39$ |
| Stand | $576.59 \pm 371.00$ | $918.55 \pm 29.30$ | $\mathbf{937.20 \pm 14.28}$ |
| Sit_Hard | $511.85 \pm 155.45$ | $\mathbf{756.67 \pm 108.79}$ | $678.53 \pm 106.32$ |

## B.3 OCCLUSION-HEAVY HUMANOIDBENCH TASK

To further assess MVR under severe occlusion, we evaluate the *Pole* task from HumanoidBench, where the humanoid must move forward through a dense forest of thin poles without colliding with them. In this scenario, different body parts are frequently occluded from any single viewpoint, making multi-view coverage especially important. As shown in Tab. A8, MVR substantially outperforms the task-reward baseline TQC on this occlusion-heavy locomotion task.

## B.4 WRIST-MOUNTED CAMERA VIEWS IN METAWORLD

To complement the exocentric setup used in the main MetaWorld experiments, we consider a variant that incorporates a wrist-mounted hand camera when computing VLM scores. This variant, denoted *MVR-HandCamera*, uses the same RL and reward-model hyperparameters as MVR; the only change is that videos can also be rendered from a camera rigidly attached to the end-effector to better capture near-field contacts.

Tab. A9 compares success rates for TQC, exocentric MVR, and MVR-HandCamera on the ten CW10 tasks. MVR-HandCamera achieves comparable average performance to MVR ($0.51 \pm 0.15$ vs. $0.54 \pm 0.22$), with clear gains on contact-rich tasks such as *push-wall* and *peg-unplug-side*, while

Table A8: **HumanoidBench *Pole* task under heavy occlusion.** Returns for TQC and MVR.

| Task | TQC | MVR |
|------|-----|-----|
| Pole | $601.62 \pm 192.34$ | $\mathbf{956.43 \pm 13.02}$ |

slightly underperforming on some tasks that mainly require coarse arm motion (e.g., *push-back*). These results suggest that near-field wrist views are particularly beneficial when local object–hand interactions determine success, whereas they may provide limited benefit or mild distraction on tasks dominated by global motion.

Table A9: **MetaWorld success rates with a wrist-mounted camera.** TQC uses only task rewards. MVR uses exocentric videos, while MVR-HandCamera additionally incorporates a wrist-mounted hand camera in the reward pipeline.

| Task | TQC | MVR | MVR-HandCamera |
|------|-----|-----|----------------|
| Hammer | $0.20 \pm 0.40$ | $0.20 \pm 0.24$ | $0.20 \pm 0.24$ |
| Push-Wall | $0.10 \pm 0.20$ | $0.60 \pm 0.49$ | $0.80 \pm 0.40$ |
| Faucet-Close | $1.00 \pm 0.00$ | $1.00 \pm 0.00$ | $1.00 \pm 0.00$ |
| Push-Back | $0.60 \pm 0.37$ | $0.50 \pm 0.45$ | $0.20 \pm 0.24$ |
| Stick-Pull | $0.10 \pm 0.20$ | $0.20 \pm 0.24$ | $0.00 \pm 0.00$ |
| Handle-Press-Side | $1.00 \pm 0.00$ | $1.00 \pm 0.00$ | $1.00 \pm 0.00$ |
| Push | $0.10 \pm 0.20$ | $0.10 \pm 0.20$ | $0.10 \pm 0.20$ |
| Shelf-Place | $0.00 \pm 0.00$ | $0.20 \pm 0.24$ | $0.00 \pm 0.00$ |
| Window-Close | $1.00 \pm 0.00$ | $1.00 \pm 0.00$ | $1.00 \pm 0.00$ |
| Peg-Unplug-Side | $0.30 \pm 0.40$ | $0.60 \pm 0.37$ | $0.80 \pm 0.40$ |
| Average | $0.44 \pm 0.18$ | $0.54 \pm 0.22$ | $0.51 \pm 0.15$ |

## B.5 METAWORLD TRAINING CURVES

To further examine how MVR shapes the learning dynamics, we report success rate as a function of training steps for two MetaWorld tasks where the performance gap between MVR and TQC is particularly pronounced: *push-wall* and *peg-unplug-side*. In both cases, the MVR curves lie above the TQC baseline already in the early training phase, indicating that the shaped visual reward accelerates learning rather than only improving final asymptotic performance.

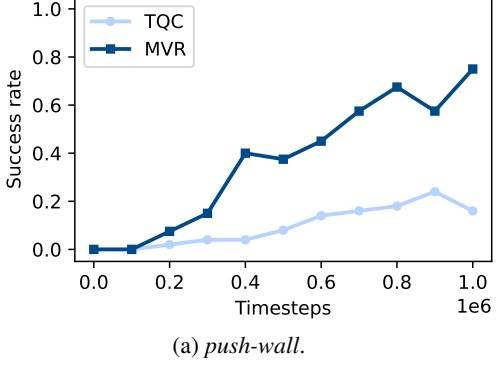 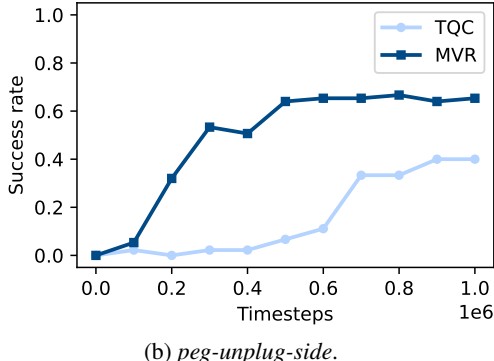

(a) *push-wall*.  (b) *peg-unplug-side*.

Figure A2: **Success rate vs. training step on two MetaWorld tasks.** Across both (a) *push-wall* and (b) *peg-unplug-side*, MVR consistently outperforms TQC throughout training, with success rates already higher in the early stages, indicating that multi-view visual shaping accelerates learning rather than only improving final performance.

Table A10: **Ablating the reward components.** Using only the VLM reward fails to solve the humanoid tasks, whereas combining it with $r^{task}$ recovers the full performance reported in Tab. 1.

| Method | Stand | Run | Slide | Sit_Hard |
|--------|-------|-----|-------|----------|
| MVR (only $r^{\text{VLM}}$) | $7.73 \pm 0.68$ | $1.51 \pm 0.05$ | $0.83 \pm 0.02$ | $2.60 \pm 0.19$ |
| MVR (full) | $918.55 \pm 29.30$ | $749.23 \pm 56.82$ | $735.03 \pm 142.85$ | $756.67 \pm 108.79$ |
| TQC (only $r^{task}$) | $576.59 \pm 371.00$ | $647.87 \pm 186.98$ | $514.91 \pm 106.36$ | $511.85 \pm 155.45$ |

## B.6 ADDITIONAL ABLATIONS

The comparison shows that relying exclusively on $r^{\text{VLM}}$ collapses learning: returns stay below ten across all four tasks. Adding the standard task reward lifts performance back to the levels reported in the main paper, highlighting that the VLM signal acts as a complement rather than a replacement for $r^{task}$.

Table A11: **Temporal averaging versus attention-based pooling.** Attention can help static tasks but remains unstable on dynamic locomotion tasks, so we adopt simple averaging in the main paper.

| Task | MVR (Temporal Avg.) | MVR (Attention) |
|------|---------------------|------------------|
| Stand | $918.55 \pm 29.30$ | $940.80 \pm 11.11$ |
| Run | $749.23 \pm 56.82$ | $519.83 \pm 366.45$ |
| Slide | $735.03 \pm 142.85$ | $683.57 \pm 243.58$ |
| Sit_Hard | $756.67 \pm 108.79$ | $822.27 \pm 18.04$ |

For the pooling study, attention improves static, single-stage behaviours such as Stand and Sit_Hard but degrades performance on locomotion tasks (Run, Slide) where the policy must react continuously. Because the weighting network observes only individual states, its outputs become noisy when the task depends on long-horizon momentum, reinforcing our choice of temporal averaging in the main ablations.

**All-view VLM ablation.** To ablate the rendering strategy, we evaluate an all-view variant, *MVR-ALL*, which renders four synchronized videos per rollout and concatenates them before querying the VLM so that every viewpoint is processed jointly. This eliminates viewpoint selection but multiplies rendering and encoding cost by four. Tab. A12 compares TQC, MVR-ALL, and our default MVR. The results are informative: MVR-ALL delivers a noticeable uplift on the static 'Stand' task yet offers only marginal or no gains on the dynamic locomotion tasks. For high-speed motions, maintaining a single coherent viewpoint preserves temporal consistency that the naive multi-view concatenation tends to disrupt, which explains why our lightweight single-view sampling coupled with relevance aggregation remains more effective despite the reduced rendering cost.

Table A12: **HumanoidBench returns for the all-view baseline.** MVR-ALL renders four videos simultaneously and feeds them jointly to the VLM. Means and standard deviations are computed over three seeds.

| Task | TQC | MVR-ALL | MVR |
|------|-----|---------|-----|
| Stand | $576.59 \pm 371.00$ | $879.06 \pm 54.27$ | $\mathbf{918.55 \pm 29.30}$ |
| Run | $647.87 \pm 186.98$ | $672.53 \pm 115.21$ | $\mathbf{749.23 \pm 56.82}$ |
| Slide | $514.91 \pm 106.36$ | $596.27 \pm 59.93$ | $\mathbf{735.03 \pm 142.85}$ |
| Sit_Hard | $511.85 \pm 155.45$ | $510.78 \pm 115.49$ | $\mathbf{756.67 \pm 108.79}$ |

## B.7 REWARD CORRELATION ANALYSIS

*Quantitative alignment.* We evaluated 100 Sit_Hard rollouts and computed the trajectory-averaged $f^{\text{VLM}}$, $f^{\text{MVR}}$, and $r^{\text{MVR}}$. As summarised in Tab. A13, their Pearson correlations with the binary success indicator reach 0.91, 0.96, and 0.98, respectively. Importantly, the relevance score $f^{\text{MVR}}$ keeps a strong link to success while remaining only weakly correlated with the sparse environment

reward (0.22), indicating that the learned shaping emphasises task completion cues rather than copying $r^{task}$. Consequently the shaped reward $r^{MVR}$ maintains the success alignment and, because it respects the ordering of successful trajectories, leaves the optimal policy approximately unchanged while reinforcing the visual priors encoded by the relevance model. These measurements explain why visual feedback is indispensable: it resolves ambiguities in $r^{task}$ while remaining tightly aligned with the actual success criterion.

*Automatic shaping.* Tracking the same 100 rollouts over training shows that the average magnitude of $r^{VLM}$ during the final 100 steps of successful episodes drops to about $23\%$ of its early-training value. This confirms empirically that $r^{MVR}$ hands control back to $r^{task}$ as soon as the desired behaviour emerges, preventing long-term gradient conflicts.

Tab. A13 summarises the correlations measured on Sit_Hard (identical to those referenced in the main text). The statistics highlight that $f^{MVR}$ remains strongly aligned with success yet only weakly correlated with $r^{task}$, while the shaped reward $r^{MVR}$ maintains the ordering of successful rollouts, keeping the optimal policy approximately unchanged.

Table A13: **Reward-signal correlations on Sit_Hard.**

| Signal | Pearson vs success | Pearson vs $r^{task}$ |
|---|---|---|
| Raw VLM similarity $f^{VLM}$ | 0.91 | 0.21 |
| Learned relevance $f^{MVR}$ | 0.96 | 0.22 |
| Shaped reward $r^{MVR}$ | 0.98 | 0.85 |

We further compare reward quality across VLM backbones. Tab. A14 reports the correlation between the raw similarity signal and $r^{task}$ together with the fraction of trajectories exhibiting negative correlation; the raw signal varies widely across ViCLIP-L and ViCLIP-B, whereas the learned relevance remains small but positive in both cases.

We also assess robustness with respect to success correlation. Tab. A15 shows that the learned relevance and shaped reward remain positive and nearly unchanged across the two backbones, so the MVR pipeline still produces a stable, task-aligned signal when the base VLM is reduced in capacity.

Table A14: **Correlation with $r^{task}$ and fraction of negatively correlated trajectories (Sit_Hard).**

| Backbone | corr($f^{VLM}, r^{task}$) | Negative traj. | corr($f^{MVR}, r^{task}$) |
|---|---|---|---|
| ViCLIP-L (428M) | 0.21 | 50% | 0.22 |
| ViCLIP-B (150M) | 0.51 | 10% | 0.23 |

Table A15: **Correlation with success across VLM backbones (Sit_Hard).**

| Backbone | corr($f^{VLM}$, success) | corr($f^{MVR}$, success) | corr($r^{MVR}$, success) |
|---|---|---|---|
| ViCLIP-L (428M) | 0.91 | 0.96 | 0.98 |
| ViCLIP-B (150M) | 0.90 | 0.95 | 0.97 |

## B.8    THE EFFECT OF WEIGHTING PARAMETER $w$

The MVR framework introduces a hyperparameter $w$ that scales the VLM-derived component in the shaped reward $r^{MVR}$ within the policy objective Eq. (6). We investigated the impact of different values of $w$ on MVR's performance, selecting $w$ from $\{0.01, 0.1, 0.5\}$ via grid search. Fig. A3 shows the final performance for different $w$ values across selected tasks.

The results suggest that the optimal balance between task rewards and VLM guidance is task-specific. Tasks involving rapid, dynamic motion (e.g., 'Run') may benefit from smaller $w$, prioritizing the optimization of the primary task objective ($r^{task}$). Tasks requiring precise posture or stability (e.g., 'Stand') might benefit more from larger $w$, leveraging the VLM's visual understanding encoded in $r^{MVR}$ to achieve the desired configuration. This highlights the need for careful tuning of $w$ according to the specific demands of the task. (Hu et al., 2020) is a potential solution for automatic parameter selection.

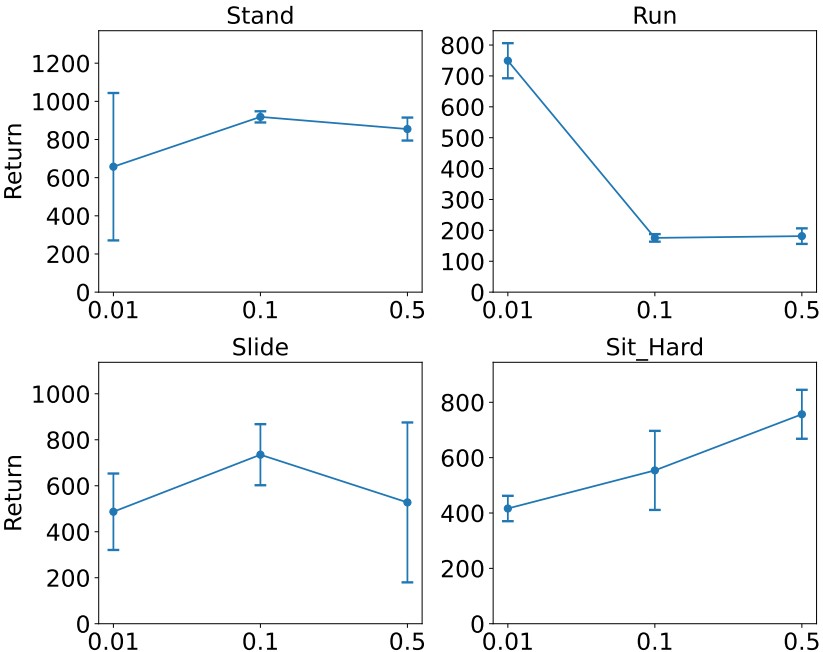

Figure A3: **The optimal value of** $w$ **in Eq. (6) is task-dependent**. Smaller values ($w = 0.01$) appear beneficial for tasks requiring rapid motion like 'Run', allowing the agent to prioritize the task reward. Conversely, tasks emphasizing stability like 'Stand' benefit from larger values ($w = 0.1$ or $w = 0.5$), giving more weight to the VLM guidance. 'Sit_Hard' shows peak performance at $w = 0.5$, while 'Slide' performs best at $w = 0.1$. This indicates that tuning $w$ based on task requirements may be necessary for optimal results.

## B.9 Comparison with FuRL

We include additional comparative results against FuRL (Fu et al., 2024), a method that also utilizes visual information for reward generation. FuRL is based on image-text similarity, and its core mechanism involves leveraging successful trajectories to perform online fine-tuning of the Vision-Language Model's (VLM) representations. This online fine-tuning aims to address potential VLM misalignment with specific task objectives. To facilitate a comparison on HumanoidBench, we operated FuRL under a VLM interaction and representation update schedule comparable to MVR's (i.e., fitting a reward model based on a similar rendering frequency), ensuring computational tractability for these demanding tasks. The results for FuRL on MetaWorld tasks, as presented in Tab. A16, are taken directly from the original FuRL publication (Fu et al., 2024). For HumanoidBench, we conducted new experiments with FuRL operating under this adjusted interaction scheme, and these results are summarized in Tab. A17.

FuRL's performance on HumanoidBench tasks (Table A17), when operated under a VLM interaction schedule adjusted for scalability, is notably lower. A core potential reason is that FuRL's strategy of fine-tuning VLM representations using successful trajectories is severely hampered in Humanoid-Bench. The inherent difficulty of exploration in these complex environments (cf. Appendix B.1) likely means that high-quality successful trajectories, which are crucial for FuRL's VLM fine-tuning process, are rarely discovered. This scarcity of effective training data would directly impede the optimization of VLM embeddings, thus limiting the quality of the visual guidance FuRL can provide for these challenging humanoid control tasks.

Table A16: **MVR demonstrates competitive or superior performance compared to FuRL on the selected MetaWorld tasks for which FuRL results are available.** Both methods combine task-specific rewards ($r^{\text{task}}$) with VLM-based rewards. FuRL results are sourced from (Fu et al., 2024).

| Task | MVR | FuRL |
|---|---|---|
| push | $\mathbf{0.10 \pm 0.20}$ | $0.06 \pm 0.08$ |
| window-close | $\mathbf{1.00 \pm 0.00}$ | $1.00 \pm 0.00$ |
| **Average** | $\mathbf{0.55 \pm 0.10}$ | $0.53 \pm 0.04$ |

Table A17: **MVR significantly outperforms FuRL on all tested dynamic and static Humanoid-Bench tasks.** For this comparison, FuRL, which combines task rewards ($r^{\text{task}}$) with VLM-based rewards, was assessed using a VLM guidance mechanism updated at a frequency similar to MVR's for scalability. These results further underscore the effectiveness of MVR's approach in leveraging multi-view video guidance from VLMs for complex humanoid control.

| Task | MVR | FuRL |
|---|---|---|
| Walk | $\mathbf{927.47 \pm 1.83}$ | $1.64 \pm 0.31$ |
| Run | $\mathbf{749.23 \pm 56.82}$ | $13.32 \pm 12.79$ |
| Stair | $\mathbf{208.60 \pm 166.22}$ | $4.41 \pm 2.32$ |
| Slide | $\mathbf{735.03 \pm 142.85}$ | $19.74 \pm 11.83$ |
| Stand | $\mathbf{918.55 \pm 29.30}$ | $30.15 \pm 8.31$ |
| Sit_Simple | $\mathbf{861.07 \pm 19.35}$ | $196.15 \pm 154.17$ |
| Sit_Hard | $\mathbf{756.67 \pm 108.79}$ | $24.56 \pm 2.69$ |
| Balance_Simple | $\mathbf{299.87 \pm 58.45}$ | $21.32 \pm 11.43$ |
| Balance_Hard | $\mathbf{95.78 \pm 11.03}$ | $15.67 \pm 3.11$ |

## B.10 Performance against Original RoboCLIP on HumanoidBench

We further benchmark MVR against the original RoboCLIP codebase (PPO, S3D) on Humanoid-Bench. This specific evaluation is critical as it assesses MVR against RoboCLIP in its unaltered, publicly available form. The substantially poorer performance of the original RoboCLIP, detailed in Table A18, likely stems from several factors inherent to its design. Firstly, its reliance on the S3D vision model, while capable of video processing, may provide less nuanced visual-semantic understanding for reward generation compared to the more advanced VLMs leveraged by MVR.

Secondly, RoboCLIP's original reward mechanism might not as effectively capture complex, temporally extended behaviors or handle viewpoint limitations as MVR's multi-view video-based approach. This contrasts with other analyses in this paper where baseline configurations (e.g., PPO to TQC, S3D to ViCLIP) were adapted for fairer component-wise comparisons, isolating specific architectural contributions.

Table A18: **MVR substantially outperforms the original RoboCLIP (PPO, S3D) across all HumanoidBench tasks.** The original RoboCLIP combines task rewards ($r^{\text{task}}$) with VLM-based rewards from its S3D model.

| Category | Task | MVR | RoboCLIP |
|---|---|---|---|
| Dynamic | Walk | $\mathbf{927.47 \pm 1.83}$ | $25.31 \pm 5.98$ |
| | Run | $\mathbf{581.33 \pm 210.35}$ | $8.19 \pm 3.09$ |
| | Stair | $\mathbf{164.74 \pm 132.96}$ | $19.25 \pm 8.63$ |
| | Slide | $\mathbf{622.05 \pm 195.11}$ | $27.37 \pm 8.28$ |
| Static | Stand | $\mathbf{925.55 \pm 25.10}$ | $36.67 \pm 7.66$ |
| | Sit_Simple | $\mathbf{757.34 \pm 143.96}$ | $16.95 \pm 1.10$ |
| | Sit_Hard | $\mathbf{705.29 \pm 105.18}$ | $11.53 \pm 4.20$ |
| | Balance_Simple | $\mathbf{284.54 \pm 46.82}$ | $45.51 \pm 1.39$ |
| | Balance_Hard | $\mathbf{95.78 \pm 11.03}$ | $36.42 \pm 1.83$ |

### B.11 PIXEL-BASED METAWORLD EXPERIMENTS

We additionally evaluate MVR on the MetaWorld suite when policies receive pixel observations. Each agent augments proprioceptive input with embeddings from the R3M-ViT encoder (Nair et al., 2022), while MVR reuses the multi-view reward pipeline without extra tuning. The shaped reward $r_t^{\text{MVR}}$ still combines environment feedback with the learned visual guidance. Results in Tab. A19 show that MVR preserves its success-rate advantage even with raw RGB inputs.

Table A19: **Pixel-conditioned MetaWorld success rates.** Both agents operate on proprioception concatenated with R3M-ViT features. Returns are averaged over five seeds.

| Task | TQC (vision) | MVR (vision) |
|---|---|---|
| hammer | $0.17 \pm 0.24$ | $\mathbf{0.80 \pm 0.28}$ |
| push-wall | $\mathbf{0.33 \pm 0.47}$ | $0.17 \pm 0.24$ |
| faucet-close | $\mathbf{1.00 \pm 0.00}$ | $\mathbf{1.00 \pm 0.00}$ |
| push-back | $0.00 \pm 0.00$ | $0.00 \pm 0.00$ |
| stick-pull | $0.00 \pm 0.00$ | $0.00 \pm 0.00$ |
| handle-press-side | $\mathbf{1.00 \pm 0.00}$ | $\mathbf{1.00 \pm 0.00}$ |
| push | $0.00 \pm 0.00$ | $0.00 \pm 0.00$ |
| shelf-place | $0.00 \pm 0.00$ | $0.00 \pm 0.00$ |
| window-close | $\mathbf{1.00 \pm 0.00}$ | $\mathbf{1.00 \pm 0.00}$ |
| peg-unplug-side | $0.00 \pm 0.00$ | $\mathbf{0.20 \pm 0.22}$ |
| Average | $0.35 \pm 0.07$ | $\mathbf{0.42 \pm 0.07}$ |

### B.12 SPARSE-REWARD METAWORLD EXPERIMENTS

To further assess MVR in settings where dense task rewards are unavailable, we construct a sparse-reward variant of the MetaWorld benchmark. In this setting, we remove all dense environment rewards and keep only the binary success signal, while leaving the visual shaping signal $r^{\text{VLM}}$ unchanged. We then compare three configurations: TQC with dense rewards only (TQC-Dense), MVR with dense rewards (MVR-Dense, identical to the main-paper setting), and MVR with sparse environment rewards (MVR-Sparse).

Overall, MVR-Sparse attains an average success rate comparable to TQC-Dense while removing dense environment rewards, indicating that MVR can leverage visual shaping to compensate for the lack of hand-engineered dense task rewards.

Table A20: **Sparse-reward MetaWorld experiments.** MVR-Sparse uses only a sparse success signal from the environment plus visual shaping, yet achieves competitive performance with the dense-reward baseline TQC-Dense. Results are averaged over five seeds.

| Task | TQC-Dense | MVR-Dense | MVR-Sparse |
|---|---|---|---|
| hammer | $0.20 \pm 0.40$ | $0.20 \pm 0.24$ | $\mathbf{0.90 \pm 0.20}$ |
| push-wall | $0.10 \pm 0.20$ | $\mathbf{0.60 \pm 0.49}$ | $0.30 \pm 0.24$ |
| faucet-close | $\mathbf{1.00 \pm 0.00}$ | $\mathbf{1.00 \pm 0.00}$ | $\mathbf{1.00 \pm 0.00}$ |
| push-back | $\mathbf{0.60 \pm 0.37}$ | $0.50 \pm 0.45$ | $0.00 \pm 0.00$ |
| stick-pull | $0.10 \pm 0.20$ | $\mathbf{0.20 \pm 0.24}$ | $0.00 \pm 0.00$ |
| handle-press-side | $\mathbf{1.00 \pm 0.00}$ | $\mathbf{1.00 \pm 0.00}$ | $\mathbf{1.00 \pm 0.00}$ |
| push | $\mathbf{0.10 \pm 0.20}$ | $\mathbf{0.10 \pm 0.20}$ | $0.00 \pm 0.00$ |
| shelf-place | $0.00 \pm 0.00$ | $\mathbf{0.20 \pm 0.24}$ | $0.00 \pm 0.00$ |
| window-close | $\mathbf{1.00 \pm 0.00}$ | $\mathbf{1.00 \pm 0.00}$ | $\mathbf{1.00 \pm 0.00}$ |
| peg-unplug-side | $0.30 \pm 0.40$ | $0.60 \pm 0.37$ | $\mathbf{0.90 \pm 0.20}$ |
| Average | $0.44 \pm 0.18$ | $\mathbf{0.54 \pm 0.22}$ | $0.51 \pm 0.064$ |

## C    ADDITIONAL ILLUSTRATION

### C.1    OBJECTIVE FUNCTIONS FOR STATE RELEVANCE LEARNING

As detailed in Sec. 4.1, learning the state relevance function $f^{\mathrm{MVR}}$ involves addressing challenges arising from the discrepancy between state-space representations and visual video data, as well as viewpoint-induced variability in similarity scores when the same trajectory is rendered from different cameras. MVR tackles these using two learning objectives, $L_{\mathrm{matching}}$ and $L_{\mathrm{reg}}$, illustrated in Fig. A4.

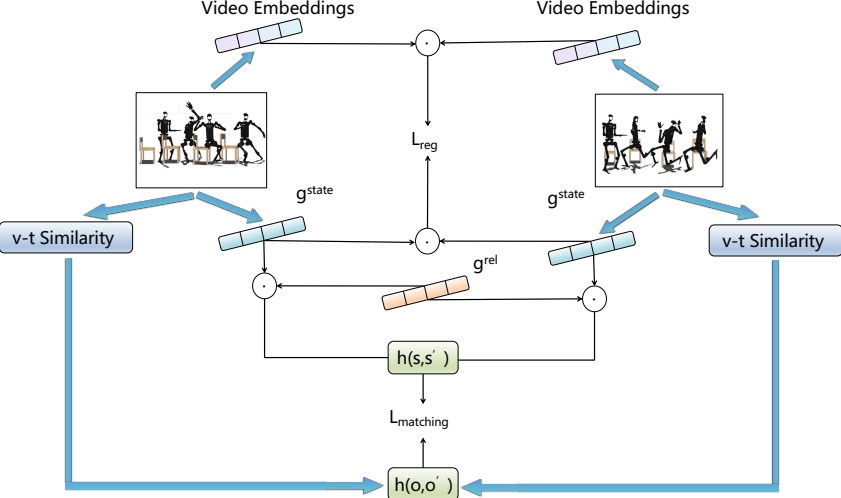

Figure A4: **The two objectives for learning state relevance**. The state relevance function $f^{\mathrm{MVR}}$ comprises a state encoder $g^{\mathrm{state}}$ and a relevance vector $g^{\mathrm{rel}}$. $L_{\mathrm{matching}}$ aligns the relative ranking of state sequences (derived from $f^{\mathrm{MVR}}$) with the ranking induced by VLM video-text similarity scores. $L_{\mathrm{reg}}$ regularizes the state sequence representations by enforcing consistency with the similarity structure observed in the VLM's video embedding space. This dual approach allows MVR to learn state relevance effectively while mitigating noise from visual details and viewpoint variations.

$L_{\mathrm{matching}}$ encourages the state relevance function to preserve the relative ordering of behaviors indicated by the VLM's assessment of corresponding videos. $L_{\mathrm{reg}}$ aligns the distance metric in the learned state embedding space with the distances in the VLM's video embedding space, promoting representations robust to view-specific interference. Together, these objectives enable MVR to learn a meaningful state relevance function from multi-view video feedback.

## D    LIMITATIONS AND FUTURE WORK

MVR currently relies on a single textual description per task, which can underspecify complex behaviours. Extending the framework with adaptive or stage-specific prompts is a promising direction for capturing richer task semantics.

The prototype assumes access to multiple calibrated cameras to gather multi-view footage. Deployments with limited viewpoints may require coupling MVR with novel-view synthesis modules such as NeRFs (Mildenhall et al., 2020) or active camera placement to maintain coverage.

While VLM guidance helps expose dynamic motion cues, combining video-text similarity with other perceptual signals may broaden applicability to tasks where text alignment alone is insufficient. Finally, automating the selection of the weighting parameter $w$ in the policy objective would reduce hyperparameter tuning effort and improve robustness across domains.

## E    BROADER IMPACTS

Robot skill learning is an important topic for robotics and embodied artificial intelligence. This paper presents an approach that solicits feedback from VLMs for skill learning, thereby leveraging the rich knowledge stored in such models. With such a method, this paper aims to pave the road towards automatic robot skill learning, a desideratum of learning systems that enables intelligent robots to acquire diverse real-world tasks.

As a foundational methodological contribution, this work does not present direct applications with immediately foreseeable negative societal impacts. Nevertheless, continued research in this foundational area, conducted with an awareness of broader ethical considerations, is crucial for building a deeper understanding and fostering the responsible development of more advanced autonomous systems in the future.

**LLM Usage.**    A general-purpose large language model (ChatGPT) was used solely for polishing grammar and improving the clarity of draft paragraphs. All technical content, equations, experimental design, and analysis were created and verified by the authors.

