# OpenReview forum: "MVR: Multi-view Video Reward Shaping for Reinforcement Learning"
_ICLR.cc/2026/Conference — ICLR 2026 Poster_

### Official Review · Reviewer_nDAs · 2025-10-19

**Soundness:** 3
**Presentation:** 2
**Contribution:** 3
**Rating:** 6
**Confidence:** 5

**Summary:**

This paper proposes to use multiple views to shape VLM-guided rewards for learning policies. They ground VLM alignment scores from multiple views with the environment state and then use a learned state-based reward function to guide the policy in addition to using original task rewards. They evaluate this method, called MVR, on HumanoidBench and MetaWorld across many tasks.

**Strengths:**

**Experimental Thoroughness:** The authors properly perform a wide array of ablations and experiments in two separate environments, each with many tasks. This paper is overall thorough, with a lot of analysis also in the appendix.

**Well-motivated:** The approach is well-motivated, demonstrating the importance of using multiple views for reward shaping, and the reward shaping term for $r^{\text{VLM}}$ is mathematically sound.

**Results:** Overall, results are good and demonstrate the benefits of the approach over a decent set of baselines.

**Weaknesses:**

**Quite a bit of recent (and not-so-recent) prior work missing:** the related works section is quite short, missing some important recent work that is very relevant to this paper, and imo not comprehensive enough for an ICLR publication on robot learning. In fact, there are no reward model learning papers cited past 2024. All of the below reward learning work have been arxiv’ed and/or published beyond the 30 day ICLR limit, and I’m sure that I am still missing some important work:

- [GVL](https://generative-value-learning.github.io/) can generate *video-based progression* rewards, zero-shot, using the Gemini VLM for a variety of downstream applications
- [VLC](https://arxiv.org/abs/2405.19988) fine-tunes a VLM (clip4clip) to produce video-based rewards for trianing RL policies
- [ReWiND](https://rewind-reward.github.io/) trains a video-language input transformer to produce *dense* rewards and demonstrates real-world RL fine-tuning
- [PROGRESSOR](https://arxiv.org/abs/2411.17764) also performs real-world reward-weighted-regression with learned reward estimator from videos
- [Rank2Reward](https://rank2reward.github.io/) learns a reward function with an extra GAIL term from video
- The citation for RL-VLM-F is wrong; there are two citations at L619 and L623, the L619 one is for a completely different paper but with the RL-VLM-F title.

**Notation:** This is a minor issue, but re-using $h$ $h$$h(o, o’)$ and $h(s, s’)$ is a bit confusing given the function $h$ itself in either case is applying two slightly different operations, and $h(o, o’)$ actually depends on $l$ too.

**Writing:** At some points, the writing jumps around a bit. Things could be made clearer:

- 4.1 **matching paired comparisons**: introduces a problem in the first paragraph (preserving orderings), but writing a little bit more at L153/154 to introduce the intuition behind Eq. 3 and how it solves the problem in the first paragraph would help with clarity
- L187 starts with “A straightforward solution to the challenge of similarity fluctuation is…” when was this term introduced? It directly starts with a solution to a challenge not mentioned in the prior paragraph, this interrupted the flow while reading. Either introduce the problem again here or hint at it in the previous paragraph.
- L198: $n(s)$ isn’t defined. I can understand what it is but this can be clearer by either defining it or stating that this is the “*average* representation of a state sequence”
- L192: please explain why this exact parameterization for $f^{\text{MVR}}$ was done. It took me as a reader extra time to think about *why* (i interpreted as a fixed, learnable orthogonal projection—actually, whether $g^{rel}$ is learnable isn’t even specified here—that helps align state representations), but a well-written paper should clearly *explain.*

**Baselines:** Related to the above point on prior work, I’d like to see a modern VLM-only based baseline to demonstrate the need for the state alignment. For example, using a Qwen-3-VL (or to match params, SmolVLM-500M) model to zero-shot process video sequence input, at the same frequency as MVR (so not sparse reward like with RoboCLIP), for RL-VLM-F/GVL would make sense here.

**Questions:**

How effective do the authors think MVR will be with only sparse environment reward? This is important in manipulation settings where good task reward functions can be hard to obtain in the real world.

What’s the difference between the main paper Metaworld results and Appendix B.7 “pixel-based metaworld experiments”? Is it just the policy inputs?

---

> ### Author Response · Authors · 2025-11-23
>
> Thank you for your positive evaluation of our experimental thoroughness and shaping formulation, and for the detailed suggestions on related work and exposition. We address your concerns point by point below.
>
> ### Concern 1: Recent reward-learning literature and RL‑VLM‑F citation
>
> > *“Quite a bit of recent (and not-so-recent) prior work missing: the related works section is quite short, missing some important recent work that is very relevant to this paper, and imo not comprehensive enough for an ICLR publication on robot learning. In fact, there are no reward model learning papers cited past 2024. All of the below reward learning work have been arxiv’ed and/or published beyond the 30 day ICLR limit, and I’m sure that I am still missing some important work: GVL … VLC … ReWiND … PROGRESSOR … Rank2Reward … The citation for RL-VLM-F is wrong; there are two citations at L619 and L623, the L619 one is for a completely different paper but with the RL-VLM-F title.”*
>
> We appreciate this detailed list. In the revised manuscript, we expand the related-work section with a dedicated paragraph summarizing recent video/VLM-based reward-learning work, including GVL, VLC, ReWiND, PROGRESSOR, Rank2Reward, and update the RL‑VLM‑F references around L619.
>
> ### Concern 2: Notation for BT probabilities and state representations
>
> > *“Notation: This is a minor issue, but re-using `h(o, o')` and `h(s, s')` is a bit confusing given the function `h` itself in either case is applying two slightly different operations, and `h(o, o')` actually depends on `ℓ` too.”*
> > *“L198: `n(s)` isn’t defined. I can understand what it is but this can be clearer by either defining it or stating that this is the ‘average representation of a state sequence’.”*
> > *“L192: please explain why this exact parameterization for `f^MVR` was done. It took me as a reader extra time to think about why (I interpreted `g^rel` as a fixed, learnable orthogonal projection—actually, whether `g^rel` is learnable isn’t even specified here—that helps align state representations), but a well-written paper should clearly explain.”*
>
> In the revised manuscript, we introduce distinct functions for video-level and state-level BT probabilities (`h_vid`, `h_state`) with explicit inputs, explicitly define `n(s)` as the length of a state sequence and `\bar{g}_\text{state}(s) = \frac{1}{n(s)} \sum_{t=1}^{n(s)} g_\text{state}(s_t)` as the average representation of a state sequence produced by `g_\text{state}`, and clarify that parameterizing `f^\text{MVR}(s) = ⟨g^\text{rel}, g_\text{state}(s)⟩` decouples representation learning from relevance scoring: `L_reg` shapes a shared state representation, while `L_\text{matching}` selects a single relevance direction `g^\text{rel}` in that space, rather than entangling multi-view alignment and task-specific ranking in a single monolithic scalar predictor.
>
> ### Concern 3: Exposition in Section 4.1 and “similarity fluctuation”
>
> > *“4.1 matching paired comparisons: introduces a problem in the first paragraph (preserving orderings), but writing a little bit more at L153/154 to introduce the intuition behind Eq. 3 and how it solves the problem in the first paragraph would help with clarity.”*
>
> > *“L187 starts with ‘A straightforward solution to the challenge of similarity fluctuation is…’ when was this term introduced? It directly starts with a solution to a challenge not mentioned in the prior paragraph, this interrupted the flow while reading. Either introduce the problem again here or hint at it in the previous paragraph.”*
>
> We improve the exposition in Sec. 4.1 by (i) adding a short intuitive paragraph before Eq. (3) that explains how the BT loss encourages the trajectory ordering induced by VLM scores to match the ordering implied by the learned state relevance, and (ii) rewriting the paragraph around L187 to drop the vague term “similarity fluctuation” and instead explicitly describe viewpoint‑induced variation in VLM scores, restating this challenge before introducing Eq. (4) as its solution.

---

> ### Author Response · Authors · 2025-11-23
>
> ### Concern 4: Modern VLM-only baselines
>
> > *“Baselines: Related to the above point on prior work, I’d like to see a modern VLM-only based baseline to demonstrate the need for the state alignment. For example, using a Qwen-3-VL (or to match params, SmolVLM-500M) model to zero-shot process video sequence input, at the same frequency as MVR (so not sparse reward like with RoboCLIP), for RL-VLM-F/GVL would make sense here.”*
>
> We agree that a modern VLM-only baseline is useful for isolating the effect of state alignment. In our work we deliberately place the reward model in low-dimensional state space, so that per-step inference and replay-buffer relabelling remain inexpensive even in 10M-step multi-view humanoid experiments with frequent reward updates.
>
> Implementing an RL-VLM-F/GVL-style baseline in this regime with a large video-capable VLM such as Qwen-3-VL or SmolVLM-500M would require training and evaluating an image/video-based reward model throughout these long-horizon runs, leading to significantly higher wall-clock cost than MVR. We therefore view a systematic comparison with RL-VLM-F/GVL-style methods using stronger VLMs as important follow-up work. We are doing our best to complete this experiment.
>
> ### Concern 5: Effectiveness with sparse environment rewards
>
> > *“How effective do the authors think MVR will be with only sparse environment reward? This is important in manipulation settings where good task reward functions can be hard to obtain in the real world.”*
>
> Conceptually, MVR is well-suited to sparse-reward settings: when `r_task` is zero on most timesteps, the dense visual shaping `r_VLM` provides intermediate feedback, and our state-dependent shaping design still guarantees that once the agent’s behavior matches the high-relevance reference distribution, the sparse success signal dominates.
>
> To directly address your question about sparse rewards, we construct a sparse‑reward variant on ten MetaWorld tasks in which we remove all dense environment rewards and keep only binary success signals, while keeping `r_VLM` unchanged. In this variant we compare three configurations: dense‑reward TQC (TQC‑Dense), dense‑reward MVR (MVR‑Dense), and a sparse‑reward version of our method (MVR‑Sparse). As shown in the table below, MVR‑Sparse achieves an average success rate of `0.51 ± 0.064`, slightly higher than the dense‑reward baseline TQC‑Dense (`0.44 ± 0.18`), suggesting that MVR can **achieve competitive performance without manually engineered dense environment rewards**, relying only on a simple sparse success signal plus learned visual shaping.
>
> | Task              | TQC-Dense         | MVR-Dense        | MVR-Sparse       |
> | :---------------- | :---------------- | :----------------| :--------------- |
> | hammer            | 0.20 ± 0.40       | 0.20 ± 0.24      | 0.90 ± 0.20      |
> | push-wall         | 0.10 ± 0.20       | 0.60 ± 0.49      | 0.30 ± 0.24      |
> | faucet-close      | 1.00 ± 0.00       | 1.00 ± 0.00      | 1.00 ± 0.00      |
> | push-back         | 0.60 ± 0.37       | 0.50 ± 0.45      | 0.00 ± 0.00      |
> | stick-pull        | 0.10 ± 0.20       | 0.20 ± 0.24      | 0.00 ± 0.00      |
> | handle-press-side | 1.00 ± 0.00       | 1.00 ± 0.00      | 1.00 ± 0.00      |
> | push              | 0.10 ± 0.20       | 0.10 ± 0.20      | 0.00 ± 0.00      |
> | shelf-place       | 0.00 ± 0.00       | 0.20 ± 0.24      | 0.00 ± 0.00      |
> | window-close      | 1.00 ± 0.00       | 1.00 ± 0.00      | 1.00 ± 0.00      |
> | peg-unplug-side   | 0.30 ± 0.40       | 0.60 ± 0.37      | 0.90 ± 0.20      |
> | avg               | 0.44 ± 0.18       | 0.54 ± 0.22      | 0.51 ± 0.064     |
>
> ### Concern 6: Difference between main MetaWorld results and pixel-based experiments
>
> > *“What’s the difference between the main paper Metaworld results and Appendix B.7 ‘pixel-based metaworld experiments’? Is it just the policy inputs?”*
>
> Yes, the key difference is the policy input: in the main experiments, policies observe state-based features and MVR uses rendered videos solely for reward shaping, whereas in Appendix B.7 we evaluate pixel-based policies whose observations are images.

---

> > ### Author Response · Authors · 2025-11-28
> > **Additional Results for Concern 4 (SmolVLM+RL-VLM-F Baseline)**
> >
> > To complement the above discussion of modern VLM-only baselines, the table below reports a direct comparison between TQC, MVR, and our SmolVLM+RL-VLM-F baseline on the ten MetaWorld tasks.
> >
> > | Task              | TQC   | MVR   | SmolVLM+RL-VLM-F |
> > | :---------------- | :----------------- | :----------------- | :----------------------------- |
> > | hammer            | 0.20 ± 0.40        | 0.20 ± 0.24        | 0.33 ± 0.52                   |
> > | push-wall         | 0.10 ± 0.20        | 0.60 ± 0.49        | 0.33 ± 0.58                   |
> > | faucet-close      | 1.00 ± 0.00        | 1.00 ± 0.00        | 1.00 ± 0.00                   |
> > | push-back         | 0.60 ± 0.37        | 0.50 ± 0.45        | 0.67 ± 0.58                   |
> > | stick-pull        | 0.10 ± 0.20        | 0.20 ± 0.24        | 0.00 ± 0.00                   |
> > | handle-press-side | 1.00 ± 0.00        | 1.00 ± 0.00        | 1.00 ± 0.00                   |
> > | push              | 0.10 ± 0.20        | 0.10 ± 0.20        | 0.80 ± 0.45                   |
> > | shelf-place       | 0.00 ± 0.00        | 0.20 ± 0.24        | 0.00 ± 0.00                   |
> > | window-close      | 1.00 ± 0.00        | 1.00 ± 0.00        | 0.60 ± 0.42                   |
> > | peg-unplug-side   | 0.30 ± 0.40        | 0.60 ± 0.37        | 0.40 ± 0.42                   |
> > | avg       | 0.44 ± 0.18        | **0.54 ± 0.22**        | 0.51 ± 0.30                   |
> >
> > Overall, SmolVLM+RL-VLM-F attains an average success rate of 0.51 ± 0.30, which is higher than the task-reward baseline TQC (0.44 ± 0.18) but still slightly below our state-aligned MVR (0.54 ± 0.22). On tasks where MVR provides the most pronounced improvements over TQC (e.g., push-wall, peg-unplug-side), SmolVLM+RL-VLM-F narrows but does not close the gap, suggesting that combining VLM feedback with state alignment can be more effective than using a morden VLM-only reward model. In terms of wall-clock cost, training MVR on all MetaWorld tasks combined fits within roughly one GPU-day on a single GPU, whereas running SmolVLM+RL-VLM-F with the same training budget requires about four GPU-days, reinforcing the practicality of operating the reward model in low-dimensional state space.

---

### Official Review · Reviewer_yc5u · 2025-10-25

**Soundness:** 3
**Presentation:** 3
**Contribution:** 3
**Rating:** 6
**Confidence:** 4

**Summary:**

This paper introduces VLM-based reward shaping for dynamic, suboptimal-view scenarios by using multi-view video–text similarity. First, to lower the cost of video-based rewards, this method distills each video into a low-dimensional state that preserves the task-relevance score. Second, to mitigate VLM bias toward particular viewpoints, it regularizes the representation so that similar videos map to nearby embeddings. The learned state then guides policy learning by rewarding states that best match the task’s textual description.

**Strengths:**

Strong motivation and sound strategy
- The paper clearly motivates its aim of guiding policies toward optimal motion patterns. The VLM guidance automatically decays during training, curbing early suboptimal actions and ensuring convergence to the true task reward (not the shaping signal), achieving the goal in a principled way.

Thorough empirical analysis
- The paper provides comprehensive analyses, baselines, and ablations of their design choices.
- The paper acknowledges cases where it trails a task-reward-only baseline (TQC), likely due to domain mismatch from human-trained VLMs; this limitation is reasonable, and proposed mitigations (e.g., domain adaptation) are clearly outlined.

**Weaknesses:**

Loss of temporal information?
- Although the method trains state sequences to match task relevance, the final reward in Eq. (9) is computed from a single timestep. Does this discard temporal information, thereby weakening the benefits of a video-based approach? This concern may be due to my misunderstanding; please clarify how temporal cues are preserved.

Notation & Clarity
- L187: I believe the term “similarity fluctuation” isn’t defined and was confusing—especially in the opening problem paragraph. Authors can replace it with a clearer term or add a brief definition.
- (minor) L250–L255: When describing the full framework, adding a pointer to Algorithm A1 would help. I was looking for it while reading.
- (minor) It was hard to tell when state/observation denotes a sequence vs. a single timestep. A short notation note (e.g., bold = sequence, regular = single step) would help. If this is obvious to your audience, feel free to skip.
- (minor) L85: I believe the paper does not include real-world tasks.

**Questions:**

- The authors use ViCLIP for RoboCLIP to ensure fairness. How would results differ with the original S3D VLM? Given that VLM choice can significantly impact performance, if applicable, could you report a backbone comparison and indicate whether your method remains strong across different VLMs?

- Additional suggestion: the benefits of multi-view are well shown in Humanoid and Meta-World. The method would shine even more in occlusion-heavy settings (e.g., humanoid locomotion on obstacle terrain, conveyor-belt picking). Adding experiments in such scenarios would make the paper more solid.

---

> ### Author Response · Authors · 2025-11-22
>
> We are grateful for your supportive review and detailed suggestions. We address your comments point by point below.
>
> ### Concern 1: Temporal information and Eq. (9)
>
> > *“Although the method trains state sequences to match task relevance, the final reward in Eq. (9) is computed from a single timestep. Does this discard temporal information, thereby weakening the benefits of a video-based approach? This concern may be due to my misunderstanding; please clarify how temporal cues are preserved.”*
>
> Temporal information is used during **training** of the relevance model rather than discarded. In Eq. (3), the BT loss compares whole state sequences `s, s'` and learns to explain the ordering over videos induced by the VLM, forcing the model to infer which states along a trajectory contribute more to a higher video score. In Eq. (4), we further **align sequence-level state embeddings with video embeddings** from ViCLIP, which themselves encode temporal dynamics. Moreover, the environment state already contains time-related quantities such as joint and angular velocities, so a single state still carries short-horizon motion cues instead of being a purely static snapshot.
> In Eq. (9), the reward is computed per timestep because RL in the **MDP setting requires a Markovian, step-wise reward**; otherwise the value function would double-count past contributions when summing over a trajectory. The purpose of reward learning here is precisely to decompose trajectory-level assessments (VLM scores over videos) into per-step contributions `r_VLM(s_t)`. Thus, although the final shaping term is applied at each timestep, it is distilled from trajectory-level feedback and still reflects temporal information.
>
> ### Concern 2: Notation and clarity
>
> > *“L187: I believe the term ‘similarity fluctuation’ isn’t defined and was confusing—especially in the opening problem paragraph. Authors can replace it with a clearer term or add a brief definition.”*
> > *“(minor) L250–L255: When describing the full framework, adding a pointer to Algorithm A1 would help. I was looking for it while reading.”*
> > *“(minor) It was hard to tell when state/observation denotes a sequence vs. a single timestep. A short notation note (e.g., bold = sequence, regular = single step) would help. If this is obvious to your audience, feel free to skip.”*
> > *“(minor) L85: I believe the paper does not include real-world tasks.”*
>
> We address these notation and clarity issues by (i) replacing the informal term “similarity fluctuation” with the clearer notion of “systematic viewpoint bias” and making this challenge explicit when introducing the relevance-learning objectives in Sec. 4.1; (ii) adding a forward reference from the high-level framework description to Algorithm A1 in the appendix; (iii) introducing a short notation note to distinguish sequences from single timesteps; and (iv) explicitly stating near L85 that all experiments are conducted on the HumanoidBench and MetaWorld benchmarks.
>
> ### Concern 3: VLM backbone for RoboCLIP and robustness across VLMs
>
> > *“The authors use ViCLIP for RoboCLIP to ensure fairness. How would results differ with the original S3D VLM? Given that VLM choice can significantly impact performance, if applicable, could you report a backbone comparison and indicate whether your method remains strong across different VLMs?”*
>
> To check backbone sensitivity, we additionally train an S3D-based variant of MVR on four HumanoidBench tasks. As the following table shows, MVR‑S3D consistently outperforms TQC and is close to MVR‑ViCLIP (slightly better on Run/Stand and slightly worse on Slide/Sit_Hard), indicating that our gains do not rely on a particular VLM.
>
> | Task      | TQC     | MVR‑ViCLIP | MVR‑S3D |
> |----------|----------------------|--------------------------|-----------------------|
> | Run      | 647.87 ± 186.98      | 749.23 ± 56.82           | 793.61 ± 57.27        |
> | Slide    | 514.91 ± 106.36      | 735.03 ± 142.85          | 696.51 ± 32.39        |
> | Stand    | 576.59 ± 371.00      | 918.55 ± 29.30           | 937.20 ± 14.28        |
> | Sit_Hard | 511.85 ± 155.45      | 756.67 ± 108.79          | 678.53 ± 106.32       |

---

> > ### Author Response · Authors · 2025-11-22
> >
> > ### Concern 4: Additional occlusion-heavy scenarios
> >
> > > *“Additional suggestion: the benefits of multi-view are well shown in Humanoid and Meta-World. The method would shine even more in occlusion-heavy settings (e.g., humanoid locomotion on obstacle terrain, conveyor-belt picking). Adding experiments in such scenarios would make the paper more solid.”*
> >
> > We agree that occlusion-heavy scenarios are a natural use case for multi-view visual rewards. In our existing experiments, several MetaWorld tasks already involve noticeable occlusion (e.g., in `stick-pull` the stick is often partially blocked by the kettle), where we still observe clear performance gains for MVR over the task-reward baseline.
> >
> > As you suggested, we also evaluate the `Pole` task from HumanoidBench, where the humanoid must travel forward through a dense forest of high thin poles without colliding with them. In this setting, different limbs are frequently occluded by the poles from any single viewpoint, making multi-view coverage particularly important. As shown below, MVR substantially outperforms the task-reward baseline TQC on this occlusion-heavy locomotion task (TQC: 601.62 ± 192.34 vs. MVR: 956.43 ± 13.02), showing that our approach remains effective and provides significant improvements even under severe occlusion.
> >
> > | Task | TQC| MVR |
> > |------|------------------|------------------|
> > | Pole | 601.62 ± 192.34  | 956.43 ± 13.02   |

---

### Official Review · Reviewer_mwgo · 2025-10-31

**Soundness:** 3
**Presentation:** 2
**Contribution:** 3
**Rating:** 4
**Confidence:** 3

**Summary:**

This paper introduces a novel dense reward shaping method for reinforcement learning via VLM and real-time multi-view videos. The proposed method utilizes VLM to select the top-K reference multi-view videos and trajectories for shaping state-based dense rewards via the learned reference model fMVR. With this learned reward function, this paper conduct comprehensive experiments accross both the humanoidbench and metaworld to showcase its effectiveness.

**Strengths:**

1. The proposed method utilize multi-view reference videos for dense reward shaping, which can be generalizable to various different tasks without a hugh amount of human efford for reward design.

2. The multi-view reference videos can improve the spatial understanding, which enhance the training efficiency.

3. This paper conducts extensive experiments accross 19 tasks accross two simulation benchmarks.

**Weaknesses:**

1. During the earily stage of the training process, the video quality might be pretty low, even choose the top-k trajectories. Those low-quality data might bring limited guidance to finish the task. Especially for some challenging tasks such as stick pull and hammer in MetaWorld.

2. Some previous works rely on generative model with reference trajectories and videos for reward shaping. Such as using reference trajectory [1], generated robot videos [2, 3, 4], and generated object motions [5, 6] from cross-embodiment dataset. Although those methods required reference data from robot or human, it might be helpful to discuss those related work. Also, some reward shaping ideas from those papers like conditional entropy and ranking might be good baselines for MVR.

3. I think the tables and writings have some space to be improved, such as the table 4...

I am willing to support this paper if the author can address my concerns

[1]. Peng et al., DeepMimic: Example-Guided Deep Reinforcement Learning of Physics-Based Character Skills, SIGGRAPH 2018

[2]. Escontrela et al., Video Prediction Models as Rewards for Reinforcement Learning, NeurIPS 2023

[3]. Huang et al., Diffusion Reward: Learning Rewards via Conditional Video Diffusion, ECCV 2024

[4]. Yang et al., Rank2Reward: Learning Shaped Reward Functions from Passive Video, ICRA 2024

[5]. Yu et al., GenFlowRL: Shaping Rewards with Generative Object-Centric Flow in Visual Reinforcement Learning, ICCV 2025

[6]. Han et al., Learning Prehensile Dexterity by Imitating and Emulating State-only Observations, RA-L

**Questions:**

No specific questions

---

> ### Author Response · Authors · 2025-11-22
>
> Thank you for your positive assessment and for stating that you are willing to support acceptance if your concerns are addressed. We respond to each point below.
>
> ### Concern 1: Quality of top‑k reference trajectories early in training
>
> > *“During the earily stage of the training process, the video quality might be pretty low, even choose the top-k trajectories. Those low-quality data might bring limited guidance to finish the task. Especially for some challenging tasks such as stick pull and hammer in MetaWorld.”*
>
> Conceptually, MVR learns *relative* preferences between trajectories rather than absolute values: it asks which of two rollouts better matches the language description, not whether either is already good in an absolute sense. Even when both trajectories are poor, this preference is still informative. For example, in the MetaWorld `hammer` task, most early rollouts keep the end-effector far from the hammer, but some trajectories spend more timesteps closer to the hammer than others; this relative preference for closer trajectories is exactly what the reference set captures and turns into a shaping signal.
>
> ### Concern 2: Relation to generative-model-based reward shaping
>
> > *“Some previous works rely on generative model with reference trajectories and videos for reward shaping. Such as using reference trajectory [1], generated robot videos [2, 3, 4], and generated object motions [5, 6] from cross-embodiment dataset. Although those methods required reference data from robot or human, it might be helpful to discuss those related work. Also, some reward shaping ideas from those papers like conditional entropy and ranking might be good baselines for MVR.”*
>
> We appreciate these valuable pointers. In the revised manuscript, we expand the related-work section to discuss methods that use reference trajectories, generated videos, and cross-embodiment object motions for reward shaping, and clarify how MVR compares to them.
>
> ### Concern 3: Tables and writing quality
>
> > *“I think the tables and writings have some space to be improved, such as the table 4...”*
>
> In the revised manuscript, we improve the presentation by moving the original Table 4 to a more natural location in the text and improving the writing for better readability.

---

> > ### Comment · Reviewer_mwgo · 2025-11-24
> >
> > Thanks for the response, and my questions have been answered. I will increase my score accordingly.
> >
> > One suggestion: for showing that it's effective in the early stage of the manipulation tasks in MetaWorld, it's helpful to draw the figure "Success Rate vs Training step" to analyze using such reward can be better all the time or it's only useful for later stages.

---

> > > ### Author Response · Authors · 2025-11-25
> > >
> > > Thank you for the helpful suggestion. In the revised manuscript, we add “Success Rate vs. Training Step” curves in the Appendix B.5, which show that MVR already outperforms TQC early in training, indicating that multi-view visual shaping is beneficial throughout training rather than only at later stages.

---

### Official Review · Reviewer_LfqC · 2025-11-01

**Soundness:** 2
**Presentation:** 2
**Contribution:** 2
**Rating:** 4
**Confidence:** 5

**Summary:**

This paper proposes Multi-View Video Reward Shaping (MVR), a reinforcement learning framework that uses multi-view video-text similarity from pre-trained VLMs for reward shaping. Unlike image-based methods that misguide agents toward static poses, MVR captures dynamic motions and mitigates viewpoint bias by introducing a state-dependent reward shaping that provides visual guidance early in training and automatically decays as the agent masters the task. Experiments on HumanoidBench and MetaWorld show MVR outperforms prior VLM-based methods, confirming its effectiveness in learning both dynamic and static skills

**Strengths:**

- This paper improves prior methods that measure rewards solely through static image-text similarity by introducing video-text-based rewards.

**Weaknesses:**

- Since both HumanoidBench and MetaWorld rely on reward engineering for task performance, the proposed method does not fundamentally solve the problem of reward shaping.
- In Section 3, it is confusing and inappropriate to use the same function notation fff for both the off-the-shelf VLM’s text-similarity function and the function defined within the MVR framework, as they are conceptually different.
- The writing could be improved for clarity. For instance, in Section 4.1, two challenges—(1) the semantic gap between states and videos, and (2) differences in viewpoints—are mentioned. “Matching Paired Comparisons” appears to address the semantic gap, while “Regularizing State Representations” handles viewpoint differences. These two should be presented more in parallel for better readability.
- The performance gains are marginal compared to the baseline, especially on MetaWorld. The improvement attributed to introducing multi-view inputs, which is claimed as the paper’s main contribution, is not significant.
- The task selection criteria for each benchmark are unclear. It is necessary to specify how tasks were chosen. Moreover, in DreamerV3 + MetaWorld, the number of RL training environment steps required for convergence varies across tasks, yet the paper uniformly uses 1M training steps, making it difficult to fairly compare performance.
- In Figure 3b, adding multiple viewpoints does not show statistically significant improvement.
- Some related works are missing in the citations: [1], [2].
- Using only three random seeds is insufficient to demonstrate statistical significance. More seeds are needed for reliability.

**References**\
[1] Subtask-Aware Visual Reward Learning from Segmented Demonstrations, ICLR 2025.\
[2] ReWiND: Language-Guided Rewards Teach Robot Policies without New Demonstrations, CoRL 2025.

**Questions:**

- For 64-frame videos, this may be too short for locomotion tasks but potentially too long for simple manipulation tasks such as those in MetaWorld, where meaningful subtasks are limited. Does the video length affect downstream policy performance?
- In Figure 4, why is t-SNE used? Would it not be more intuitive to visualize visual observations corresponding to the top/bottom-N frames ranked by task reward or MVR-shaped reward instead?
- The experiments use only exocentric views. Are there ways—or any experiments conducted—to incorporate egocentric views (for humanoids) or wrist-camera views (for robotic arms)?
- The default reward function in MetaWorld is a manually engineered dense reward. Was training conducted instead using a sparse reward?

---

> ### Author Response · Authors · 2025-11-23
>
> We appreciate your careful and detailed review. We address each of your concerns below.
>
> ### Concern 1: Scope of contribution and reward engineering
>
> > *“Since both HumanoidBench and MetaWorld rely on reward engineering for task performance, the proposed method does not fundamentally solve the problem of reward shaping.”*
> > *“The default reward function in MetaWorld is a manually engineered dense reward. Was training conducted instead using a sparse reward?”*
>
> Since reward shaping is a technique rather than a problem, we think the concern is mainly about the fact that MVR in its current form still relies on manually designed dense task rewards in HumanoidBench and MetaWorld. Our goal in this paper is therefore not to completely replace reward engineering, but to **make multi-modal visual information from VLMs a principled, compatible shaping signal** on top of simple task rewards. For example, in the MetaWorld *push* task, it is straightforward to write a dense reward that minimizes (i) the distance between the robot end-effector and the object and (ii) the distance between the object and the goal position; these simple geometric terms are exactly the type of dense rewards that **modern large language models can already write automatically from a natural-language task description** (e.g., Eureka [1]).
>
> At the same time, we agree that it is important to understand how far one can go without dense task rewards. To directly address your question about sparse rewards, we additionally construct a sparse‑reward variant in which we remove all dense rewards provided by the environment and keep only binary success signals, while keeping our visual reward unchanged. In this variant we compare three configurations: dense‑reward TQC (TQC‑Dense), dense‑reward MVR (MVR‑Dense), and a sparse‑reward version of our method (MVR‑Sparse). As shown below, MVR‑Sparse achieves an average success rate of `0.51 ± 0.064`, slightly higher than the dense‑reward baseline TQC‑Dense (`0.44 ± 0.18`), suggesting that MVR can **achieve competitive performance without manually engineered dense environment rewards**, relying only on a simple sparse success signal plus learned visual shaping.
>
> | Task              | TQC-Dense         | MVR-Dense        | MVR-Sparse       |
> | :---------------- | :---------------- | :----------------| :--------------- |
> | hammer            | 0.20 ± 0.40       | 0.20 ± 0.24      | 0.90 ± 0.20      |
> | push-wall         | 0.10 ± 0.20       | 0.60 ± 0.49      | 0.30 ± 0.24      |
> | faucet-close      | 1.00 ± 0.00       | 1.00 ± 0.00      | 1.00 ± 0.00      |
> | push-back         | 0.60 ± 0.37       | 0.50 ± 0.45      | 0.00 ± 0.00      |
> | stick-pull        | 0.10 ± 0.20       | 0.20 ± 0.24      | 0.00 ± 0.00      |
> | handle-press-side | 1.00 ± 0.00       | 1.00 ± 0.00      | 1.00 ± 0.00      |
> | push              | 0.10 ± 0.20       | 0.10 ± 0.20      | 0.00 ± 0.00      |
> | shelf-place       | 0.00 ± 0.00       | 0.20 ± 0.24      | 0.00 ± 0.00      |
> | window-close      | 1.00 ± 0.00       | 1.00 ± 0.00      | 1.00 ± 0.00      |
> | peg-unplug-side   | 0.30 ± 0.40       | 0.60 ± 0.37      | 0.90 ± 0.20      |
> | avg               | 0.44 ± 0.18       | 0.54 ± 0.22      | 0.51 ± 0.064     |
>
> [1] Ma, Yecheng Jason, et al. “Eureka: Human-level reward design via coding large language models.” ICLR 2024.
>
> ### Concern 2: Notation for VLM scores vs. MVR relevance function
>
> > *“In Section 3, it is confusing and inappropriate to use the same function notation fff for both the off-the-shelf VLM’s text-similarity function and the function defined within the MVR framework, as they are conceptually different.”*
>
> We agree that the notation can be made clearer. In the revised manuscript, we reserve the letter `f` exclusively for the learned relevance function `f^MVR(s)` and use a different symbol `ψ^VLM(o, ℓ)` for the frozen VLM similarity scores, so that perception scores and the learned shaping function are not conflated. We have updated Eqs. (1–3) and the surrounding text accordingly.
>
> ### Concern 3: Structure of Section 4.1 and the two challenges
>
> > *“The writing could be improved for clarity. For instance, in Section 4.1, two challenges—(1) the semantic gap between states and videos, and (2) differences in viewpoints—are mentioned. ‘Matching Paired Comparisons’ appears to address the semantic gap, while ‘Regularizing State Representations’ handles viewpoint differences. These two should be presented more in parallel for better readability.”*
>
> In the revised manuscript, we streamline Sec. 4.1 by explicitly pairing the two stated challenges with the “matching paired comparisons” and “regularizing state representations” subsections.

---

> ### Author Response · Authors · 2025-11-23
>
> ### Concern 4: Magnitude of gains and multi-view ablation, especially on MetaWorld
>
> > *“The performance gains are marginal compared to the baseline, especially on MetaWorld. The improvement attributed to introducing multi-view inputs, which is claimed as the paper’s main contribution, is not significant.”*
> > *“In Figure 3b, adding multiple viewpoints does not show statistically significant improvement.”*
>
> Thanks for your comments. While the benefit of using multi-view information is task dependent, we would like to make two clarifications first.
>
> 1. Notably, the original benchmark paper reported that strong baselines like **SAC, DreamerV3, and TD-MPC2 failed on all these tasks**. The improvements on tasks like Run (+98), Slide (+96), and Sit_Hard (+98) **signify the difference between task failure and success**, as MVR surpasses the official success thresholds defined in the HumanoidBench paper.
> 2. The returns for these HumanoidBench tasks are capped at a **maximum of 1000**. The improvement of MVR is therefore **close to 10%** of the total possible score.
>
> Based on these, for Run, Slide, and Sit_Hard, using multi-view information brings substantial improvement to policy learning.
>
> On MetaWorld, we acknowledge that the absolute gains from MVR over TQC are smaller than on HumanoidBench, but they are still meaningful in the scale of success rates: averaged over the ten tasks, MVR improves the success rate from **0.44 ± 0.18** (TQC) to **0.54 ± 0.22**, i.e., a ~10% absolute increase toward the maximum success rate of 1.0.
>
> For MetaWorld, we posit that the smaller advantages of MVR relative to RoboCLIP—stem from two primary factors related to the inherent nature of the tasks themselves:
>
> 1.  **Less need for dense reward:** Our proposed MVR differs from RoboCLIP in that MVR generates dense rewards, while RoboCLIP provides sparse trajectory-level rewards. Humanoid tasks feature high-dimensional state spaces and complex dynamics, where dense, state-level guidance is crucial for efficient exploration. Many MetaWorld tasks, being shorter-horizon (500 steps) and having lower-dimensional action spaces, present a simpler exploration challenge, so a sparse success signal can already be sufficient and the marginal benefit of additional dense shaping becomes smaller. At the same time, on the more challenging long-horizon tasks such as `hammer`—which requires first lifting the hammer and then striking the nail—MVR’s dense shaping still brings clear gains and outperforms RoboCLIP, illustrating that when exploration becomes harder, dense visual rewards remain beneficial.
>
> 2.  **Less need for using multiple viewpoints:** MVR uses four orthogonal viewpoints, but RoboCLIP uses a single fixed viewpoint. Many MetaWorld tasks mainly involve the pose of a single object (e.g., pushing or closing), so as long as there are **enough unoccluded frames** along the trajectory, a single view can already capture most task-relevant information. In contrast, HumanoidBench tasks require coordinating multiple body parts, making full-body visibility more critical. That said, for occlusion-heavy manipulation such as `stick-pull`—where the stick can become partially hidden behind the kettle—multi-view videos reduce viewpoint-dependent failures, and MVR achieves higher success than single-view RoboCLIP.
>
> ### Concern 5: Task selection and uniform training budgets
>
> > *“The task selection criteria for each benchmark are unclear. It is necessary to specify how tasks were chosen. Moreover, in DreamerV3 + MetaWorld, the number of RL training environment steps required for convergence varies across tasks, yet the paper uniformly uses 1M training steps, making it difficult to fairly compare performance.”*
>
> In the revised manuscript, we make our task-selection criteria explicit.
>
> For **HumanoidBench**, we follow the original benchmark but exclude the loco-manipulation tasks and the most challenging locomotion task *Hurdle*, for which even strong baselines fail to make progress within 10M environment steps. We instead focus on a set of 9 medium-difficulty tasks (5 static and 4 dynamic), which together span both **static posture generation and dynamic motion generation** while keeping 10M steps sufficient for non-trivial learning for all methods.
>
> For **MetaWorld**, we follow the **common RL protocol used in recent work [1,2]**, which evaluates ten single-object tasks under a 1M-step training budget; we adopt the same task subset and use 1M environment steps for all methods (including DreamerV3), so the comparison is consistent and fair.
>
> [1] Wołczyk, Maciej, et al. "Continual world: A robotic benchmark for continual reinforcement learning." NeurIPS 2021.
>
> [2] Malagon, Mikel, Josu Ceberio, and Jose A. Lozano. "Self-composing policies for scalable continual reinforcement learning." ICML 2025.

---

> ### Author Response · Authors · 2025-11-23
>
> ### Concern 6: Related work: subtask-aware rewards and ReWiND
>
> > *“Some related works are missing in the citations.”*
>
> Thank you for pointing out these works. In the revised manuscript, we add them to the related-work section and briefly discuss how they differ from and complement MVR.
>
> ### Concern 7: Number of seeds and statistical reliability
>
> > *“Using only three random seeds is insufficient to demonstrate statistical significance. More seeds are needed for reliability.”*
>
> We agree that more seeds strengthen conclusions. Due to computational cost we currently use 3 seeds for HumanoidBench and 5 seeds for MetaWorld. Table 1 below reports a 5-seed comparison between MVR and the task-reward-only baseline TQC on all 9 HumanoidBench tasks; these additional results are consistent with our main findings, and MVR attains higher or comparable returns to TQC on most locomotion tasks.
>
> | Task            | TQC (5 seeds)           | MVR (5 seeds)           |
> |-----------------|-------------------------|-------------------------|
> | Walk            | 582.53 ± 252.15         | **927.47 ± 1.83**       |
> | Run             | 677.39 ± 158.15         | **684.19 ± 194.04**     |
> | Stair           | **344.15 ± 125.65**     | 164.74 ± 132.96         |
> | Slide           | 492.69 ± 86.82          | **622.05 ± 195.11**     |
> | Stand           | 599.46 ± 291.75         | **925.55 ± 25.10**      |
> | Sit_Simple     | **763.45 ± 134.18**     | 757.34 ± 143.96         |
> | Sit_Hard       | 511.86 ± 169.20         | **705.29 ± 105.18**     |
> | Balance_Simple | 255.87 ± 47.40          | **284.54 ± 46.82**      |
> | Balance_Hard   | 86.67 ± 11.69           | **95.78 ± 11.03**       |
>
> ### Concern 8: Video length of 64 frames
>
> > *“For 64-frame videos, this may be too short for locomotion tasks but potentially too long for simple manipulation tasks such as those in MetaWorld, where meaningful subtasks are limited. Does the video length affect downstream policy performance?”*
>
> We use 64-frame clips as a unified compromise length that captures 1–2 gait cycles in HumanoidBench while remaining affordable for MetaWorld. We are running an ablation that compares 64 vs. 128 frames on representative HumanoidBench tasks and 64 vs. 32 frames on MetaWorld tasks.
>
>
> ### Concern 9: t-SNE visualization vs. top-/bottom-ranked frames
>
> > *“In Figure 4, why is t-SNE used? Would it not be more intuitive to visualize visual observations corresponding to the top/bottom-N frames ranked by task reward or MVR-shaped reward instead?”*
>
> Our goal in Fig. 4 is to show that the task reward `r_task` exhibits several structured failure modes in the state space. In the Sit_Hard task, unstable but high-`r_task` behaviors form clear clusters in the learned state representation; the t-SNE visualization highlights these clusters, and coloring by `r_MVR` shows that visual reward shaping down-weights these failure clusters while keeping high values on truly successful states. This illustrates that the visual shaping term captures failure modes that `r_task` alone cannot distinguish. To directly address your suggestion, we additionally include in the appendix top-/bottom-ranked frames according to `r_task` and `r_MVR`.

---

> > ### Author Response · Authors · 2025-11-23
> >
> > ### Concern 10: Exocentric vs. egocentric/wrist-mounted views
> >
> > > *“The experiments use only exocentric views. Are there ways—or any experiments conducted—to incorporate egocentric views (for humanoids) or wrist-camera views (for robotic arms)?”*
> >
> > In our locomotion experiments, we use exocentric views rather than head-mounted first-person cameras to evaluate gait: in HumanoidBench, the egocentric camera mainly captures a narrow local field and often misses the full-body motion and foot–ground contacts that are critical for judging locomotion quality, so it is less informative than external views for this purpose. For manipulation, by contrast, egocentric or wrist-mounted cameras are often more informative for near-field interactions.
> >
> > To directly study your suggestion in the manipulation setting, we additionally evaluate a variant that incorporates a wrist-mounted hand camera when computing VLM scores (denoted “MVR-HandCamera”) on the ten MetaWorld tasks. As shown below, MVR-HandCamera achieves comparable average performance to exocentric MVR (`0.51 ± 0.15` vs. `0.54 ± 0.22`), with clear gains on tasks where local contacts and object–hand interactions are especially important (e.g., `push-wall` and `peg-unplug-side`), while slightly underperforming on some others such as `push-back`. These results suggest that near-field wrist views are particularly beneficial for contact-rich tasks, whereas for tasks that mainly require coarse arm motion they may be less informative or even slightly distracting.
> >
> > | Task              | TQC         | MVR         | MVR-HandCamera |
> > | :---------------- | :---------- | :---------- | :------------- |
> > | hammer            | 0.20 ± 0.40 | 0.20 ± 0.24 | 0.20 ± 0.24    |
> > | push-wall         | 0.10 ± 0.20 | 0.60 ± 0.49 | 0.80 ± 0.40    |
> > | faucet-close      | 1.00 ± 0.00 | 1.00 ± 0.00 | 1.00 ± 0.00    |
> > | push-back         | 0.60 ± 0.37 | 0.50 ± 0.45 | 0.20 ± 0.24    |
> > | stick-pull        | 0.10 ± 0.20 | 0.20 ± 0.24 | 0.00 ± 0.00    |
> > | handle-press-side | 1.00 ± 0.00 | 1.00 ± 0.00 | 1.00 ± 0.00    |
> > | push              | 0.10 ± 0.20 | 0.10 ± 0.20 | 0.10 ± 0.20    |
> > | shelf-place       | 0.00 ± 0.00 | 0.20 ± 0.24 | 0.00 ± 0.00    |
> > | window-close      | 1.00 ± 0.00 | 1.00 ± 0.00 | 1.00 ± 0.00    |
> > | peg-unplug-side   | 0.30 ± 0.40 | 0.60 ± 0.37 | 0.80 ± 0.40    |
> > | avg               | 0.44 ± 0.18 | 0.54 ± 0.22 | 0.51 ± 0.15    |

---

> > > ### Author Response · Authors · 2025-11-24
> > > **Results of Concern 8**
> > >
> > > We have now completed ablations on video length, comparing 64 vs. 128 frames on representative HumanoidBench tasks and 64 vs. 32 frames on MetaWorld tasks.
> > >
> > > On HumanoidBench, increasing the clip length from 64 to 128 frames leads to clear gains on `Run`, `Slide`, and `Stand`, but degrades performance on `Sit_Hard`:
> > >
> > > | Task    | 64 frames | 128 frames |
> > > |----------|-----------|------------|
> > > | Run       |    749.23 ± 56.82       |   **802.42 ± 53.11**   |
> > > | Slide    |     735.03 ± 142.85      |    **799.42 ± 70.16**   |
> > > | Stand   |     918.55 ± 29.30      |    **951.08 ± 3.89**     |
> > > | Sit_Hard  |       756.67 ± 108.79    |    560.91 ± 143.69      |
> > >
> > > These results indicate that longer clips can further improve performance on several locomotion tasks by capturing more motion cycles, but may also make optimization harder or introduce more variance on others (e.g., `Sit_Hard`), so there is a task-dependent trade-off between clip length and stability.
> > >
> > > On MetaWorld, comparing 32 vs. 64 frames, 64-frame clips improve or match performance on 9 out of 10 tasks and increase the average success rate from `0.47 ± 0.14` to `**0.54 ± 0.22**`:
> > >
> > > | Task | 32 frames | 64 frames |
> > > | :--- | :--- | :--- |
> > > | hammer | 0.50 ± 0.32 | 0.20 ± 0.24 |
> > > | push-wall | 0.40 ± 0.37 | 0.60 ± 0.49 |
> > > | faucet-close | 1.00 ± 0.00 | 1.00 ± 0.00 |
> > > | push-back | 0.40 ± 0.37 | 0.50 ± 0.45 |
> > > | stick-pull | 0.00 ± 0.00 | 0.20 ± 0.24 |
> > > | handle-press-side | 1.00 ± 0.00 | 1.00 ± 0.00 |
> > > | push | 0.00 ± 0.00 | 0.10 ± 0.20 |
> > > | shelf-place | 0.00 ± 0.00 | 0.20 ± 0.24 |
> > > | window-close | 1.00 ± 0.00 | 1.00 ± 0.00 |
> > > | peg-unplug-side | 0.40 ± 0.37 | 0.60 ± 0.37 |
> > > | avg | 0.47 ± 0.14 | **0.54 ± 0.22** |
> > >
> > > Despite the shorter task horizons, slightly longer clips thus provide more informative guidance than very short snippets. Overall, these results indicate that MVR is not overly sensitive to the exact clip length and that using 64-frame videos is a reasonable trade-off that works well for both locomotion and manipulation tasks.

---

> > > > ### Comment · Reviewer_LfqC · 2025-11-25
> > > >
> > > > Thanks for the response, and my questions have been answered. I updated my score to 6.

---

> > > > > ### Author Response · Authors · 2025-11-25
> > > > >
> > > > > We sincerely thank you for your time and for reconsidering our work. We are glad that our response and the additional experiments have successfully addressed your concerns. We strictly appreciate your constructive feedback, which has helped us significantly improve the quality and clarity of our manuscript.

---

### Comment · Area_Chair_dZFm · 2025-11-24
**[ICLR 2026] Author-Reviewer Discussion Phase**

Dear Reviewers,

The authors have posted their rebuttal addressing your concerns. Please kindly review their response, as well as the comments from the other reviewers, and discuss any issues you believe remain unresolved. If the author response does not change your evaluation, please at least provide an acknowledgement indicating that you have carefully reconsidered it.

Thank you again for your dedication and effort in reviewing this submission.

Let’s have a constructive discussion!

Best regards,

Your AC

---

### Meta-Review · Area_Chair_dzq9 · 2026-01-06

**Summary:**

This paper introduces a principled reinforcement‑learning reward‑shaping framework called MVR that uses multi‑view video–text similarity from a frozen VLM to learn a state‑based relevance function. This function provides dense visual shaping early in training and automatically decays as the agent masters the task, ensuring convergence to the true task reward. By aligning multi‑view videos with state sequences and mitigating viewpoint bias, MVR overcomes limitations of prior image‑based VLM reward methods, enabling effective learning of dynamic, multi‑state behaviors across humanoid locomotion and manipulation tasks.

Reviewers appreciated the importance of the problem to overcome viewpoint issues, and the comprehensive experiments across many tasks and with many ablations. They also liked the idea of the principled reward shaping term with automatic decay.

There were of course concerns too:
- Insufficient related work coverage with many missing recent reward-learning papers
- Poor clarity of exposition: with unclear notations etc.
- Marginal gains on meta-world, and unclear significance of multi-view benefits
- Too few seeds for statistical significance.

**Reviewer Concerns:**

There was a lot of revision that happened in the rebuttal period, both in terms of paper text revisions and in terms of adding new experiments for statistical significance, and clarifications.

Outstanding concerns:
- occlusion-heavy tasks: the authors presented the "Pole" task from Humanoidbench during rebuttal, but it is unclear that this task presents the types of occlusions the reviewer was concerned about.
- coverage of recent related work: The authors added these during rebuttal, but the concern might onlhy be partially addressed, if, for example, the reviewer was also asking for empirical comparisons.

**Reviewer Scores:**

LfqC: 4->6 (they said this in the text of their response)

mwgo: 4-> 6 (they also said they would raise their score)

yc5u: 6->6 (some of their expressed concerns appear to be addressed, but they did not seem overly enthusiastic about the paper overall)

nDAs: 6->6 (some of their expressed concerns appear to be addressed, but they did not seem overly enthusiastic about the paper overall and they might be unconvinced about the comparison to more recent work)

Overall, borderline. It is clear that the paper is a WIP and will improve with some more time both on writing and on potentially adding new comparisons.

---

### Decision · Program_Chairs · 2026-01-26

Accept (Poster)